# Upregulated hepatic lipogenesis from dietary sugars in response to low palmitate feeding supplies brain palmitate

Mackenzie E. Smith [1], Chuck T. Chen [1], Chiraag A. Gohel [2], Giulia Cisbani[1], Daniel K. Chen[1], Kimia Rezaei[1], Andrew McCutcheon [1] & Richard P. Bazinet[1] ✉

Palmitic acid (PAM) can be provided in the diet or synthesized via de novo lipogenesis (DNL), primarily, from glucose. Preclinical work on the origin of brain PAM during development is scarce and contrasts results in adults. In this work, we use naturally occurring carbon isotope ratios ($^{13}C/^{12}C$; $\delta^{13}C$) to uncover the origin of brain PAM at postnatal days 0, 10, 21 and 35, and RNA sequencing to identify the pathways involved in maintaining brain PAM, at day 35, in mice fed diets with low, medium, and high PAM from birth. Here we show that DNL from dietary sugars maintains the majority of brain PAM during development and is augmented in mice fed low PAM. Importantly, the upregulation of hepatic DNL genes, in response to low PAM at day 35, demonstrates the presence of a compensatory mechanism to maintain total brain PAM pools compared to the liver; suggesting the importance of brain PAM regulation.

During development the infant brain accretes saturated fatty acids in parallel with brain growth during the intrauterine period[1] and postnatally[2] to support structural lipids in brain tissue. A recent analysis of adult postmortem brains identified saturated fatty acids were the most abundant fatty acid species in the brain ($41 \pm 0.7$ mol%), with palmitic acid (16:0; PAM) representing nearly half of total brain saturates and $19 \pm 0.7$ mol% of total fatty acids[3], similar to previous reports of brain PAM levels relative to total fatty acids in control full-term infant brains[4]. Importantly, PAM plays a role in axonal myelination during the rapid brain growth spurt, maintaining membrane structures as well as protein palmitoylation and cell signaling[5,6]. Although PAM can be sourced directly from the diet or synthesized endogenously via de novo lipogenesis (DNL), the human milk-fed infants' diet is particularly unique in which roughly 10% of their energy intake comes from PAM, and greater than 70% of PAM is esterified at the sn-2 position of the human milk triacylglyceride (TAG); which improves the absorption of PAM in infants[7,8].

Despite the intended delivery of PAM to the infant and potential importance of PAM during development, results of preclinical studies investigating the origin of brain PAM during development are scarce[9,10]. Furthermore, results of preclinical studies investigating

brain PAM origin during development contrast those investigating the origin of PAM in the adult mouse brain[11–16]. More specifically, studies in artificially reared rat pups show labelled $^2$H-PAM does not enter brain tissue lipids, and the brain synthesizes PAM entirely endogenously[9,10]. In contrast, studies in adult rats and mice suggest labelled $^{14}$C-PAM enters brain tissue lipids after 24-hour feeding[13], intravenous injection[11,14,15], intracarotid injection[12] and in situ brain perfusion[16].

Compound specific isotope analysis (CSIA) can be used to study brain PAM origin without using tracers by taking advantage of naturally occurring differences in carbon isotope ratios (CIRs; $\delta^{13}C$; $^{13}C/^{12}C$) that exist within the environment based on how plants fix carbon during photosynthesis[17,18]. Namely, $^{13}C$ depletion exists in dietary PAM (C3 plant origin) and $^{13}C$ enrichment exists in dietary sugars including glucose (C4 plant origin); the latter being the primary substrate for hepatic DNL[19]. Therefore, enrichment or depletion in tissue PAM CIRs enables the investigation of PAM origin. Our group recently utilized CSIA to study brain PAM origin and found approximately 70% of the brain PAM pool is maintained by DNL from dietary sugars, 44% of which was derived from local DNL within the brain in response to feeding standard levels of dietary PAM to adult mice (8%)[20]. Furthermore, our group utilized CSIA and found 69–80% and 47–58% of the

[1]Department of Nutritional Sciences, University of Toronto, 1 King's College Circle, Toronto M5S 1A8 ON, Canada. [2]Department of Biostatistics and Bioinformatics, George Washington Univerusity, 950 New Hampshire Ave, NW, Washington, DC 20052, USA. ✉e-mail: richard.bazinet@utoronto.ca

brain PAM pool was maintained by total and local brain DNL from dietary sugars, respectively, and DNL from dietary sugars was augmented in adult mice fed low PAM (<2%) levels from birth compared to a high PAM intervention (>95%)[18]. However, brain PAM origin during development has not been investigated utilizing CSIA; a time when PAM is rapidly accreted to the brain[1,2].

Therefore, our study aimed to investigate the origin of brain PAM during development by measuring brain PAM CIRs at postnatal day (P) 0, 10, 21, and day 35 in response to levels of low (<2%), medium (~47%), and high (>95%) PAM levels fed from birth. Furthermore, our study conducted RNA sequencing on day 35 liver and brain tissue to identify genes and pathways that may be involved in maintaining brain PAM. To the best of our knowledge our diets have not been utilized during development, therefore, we also conducted maternal behaviour tests during gestation, as well as sensorimotor development tests in the pups during the first 10 days of life to ensure our diets were not impacting behaviour. We found the majority of brain PAM was maintained by DNL from dietary sugars, augmented in mice fed low PAM

from birth. Furthermore, we identified genes involved in hepatic DNL from dietary sugars were upregulated in mice fed low compared to high PAM at day 35 - a compensatory mechanism identified to supply the brain with PAM. Dietary PAM levels did not have a significant effect on maternal behaviour or pup sensorimotor development.

## Results

### Pup tissue total and separated PAM levels

Liver and brain tissue were collected from male pups at P0, P10, P21 and day 35 that consumed diets low (LP), medium (MP), or high (HP) in PAM from birth to assess if diet affected tissue levels of PAM. While there was no significant main effect of diet, and only time, on the relative percentage and concentration of male pup brain PAM (relative percentage: $p < 0.0001$; concentration: $p < 0.0001$) (Fig. 1A), there was a significant main effect of diet (relative percentage: $p < 0.0001$; concentration: $p = 0.0351$) and time (relative percentage: $p < 0.0001$; concentration: $p = 0.0024$) on total liver PAM levels (Fig. 1B). Across all diets, the relative percentage of male pup brain PAM peaked at P10,

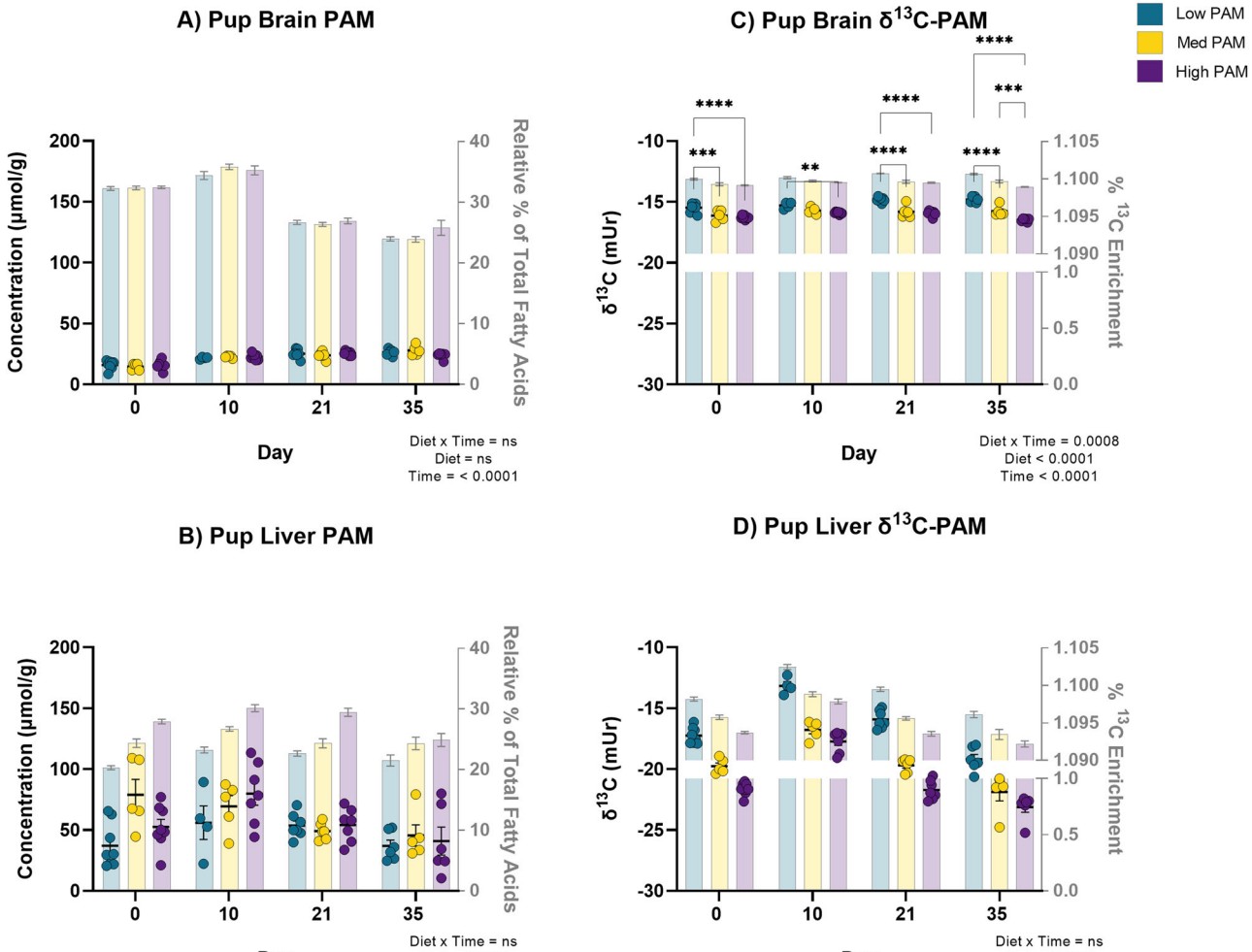

**Fig. 1 | Male pup brain palmitic acid levels are maintained compared to the liver primarily by lipogenesis from dietary sugars augmented in mice fed low palmitic acid.** While there was a significant effect of diet (relative percentage: $p < 0.0001$; concentration: $p = 0.0351$) on male pup liver palmitic acid (PAM) levels (**B**), there was only a significant main effect of time (relative percentage: $p < 0.0001$; concentration: $p < 0.0001$) on male pup brain PAM levels (**A**). Male pup brain and liver $\delta^{13}$C-PAM values were enriched overall, and there was a significant diet by time interaction on male pup brain $\delta^{13}$C-PAM (**C**) ($p = 0.0008$), and a main effect of diet ($p < 0.0001$) on male pup liver $\delta^{13}$C-PAM (**D**). Data are biological replicates presented as mean ± SEM analyzed by two-way analysis of variance (ANOVA) (**A–D**) and

Tukey's multiple comparison test (**C**); ** < 0.002, *** < 0.0002, **** < 0.0001. Brain $\delta^{13}$C-PAM means sharing a bar are significantly different by diet at each timepoint by Tukey's multiple comparison test; for all tissue $\delta^{13}$C-PAM multiple comparisons irrespective of diet and time refer to Supplementary Table 1. $n = 8, 5, 8$; $4, 5, 7$; $7, 6, 7$; $6, 5, 6$ male pups per diet per timepoint (**A–D**). Individual data points represent concentration data (for which figure statistics correspond to) while the bars show relative percentage data (**A**, **B**). Data points represent $\delta^{13}$C-PAM values (for which figure statistics correspond to) while the bars show %$^{13}$C enrichment ($^{13}$C/($^{13}$C+$^{12}$C)) (**C**, **D**). Source data are provided as a Source Data file. $\delta^{13}$C, $^{13}$C/$^{12}$C; Med medium; PAM palmitic acid.

significantly higher than at P21 and day 35 ($p < 0.0001$). As opposite, the concentration of male pup brain PAM significantly increased with time from P10 to P21 and day 35 ($p < 0.0001$).

Because total PAM levels were not different by diet in male pup brains, we explored if there may be a difference in PAM levels in individual phospholipid (PL) fractions, namely choline glycerophospholipids (ChoGpl), ethanolamine glycerophospholipids (EtnGpl), phosphatidylinositol (PtdIns), phosphatidylserine (PtdSer) and sphingomyelin (CerPCho) in a subset of male pup brains. P0 and P10 were excluded from statistical analysis due to limited brain material. Nevertheless, levels of male pup PAM in separated PL fractions at P0 and P10 do not appear to be influenced by diet (Supplementary Fig. 1A, B). Furthermore, consistent with the maintenance of total brain PAM in response to diet, at P21 and day 35 the concentration of male pup PAM levels were only significantly different by PL fraction ($p < 0.0001$ for all) (Supplementary Fig. 1C, D), as levels of PAM in different PL pools are known to be different[11,21].

In male pup liver, the difference in the level of total PAM was most pronounced between pups fed the HP and LP diets and there was a significant dose-response reduction in liver PAM, in which lower dietary PAM resulted in significantly lower liver PAM (relative percentage: $p < 0.0001$; concentration: $p = 0.00351$) (Fig. 1B). The significant dose-response reduction in the relative percentage, but not concentration, of total pup male liver PAM in response to diet persisted in a subset of separated male pup liver neutral lipids at day 35 including TAG ($p < 0.0001$) and cholesteryl ester (CE) ($p = 0.0135$) fractions (Supplementary Fig. 2A, C), however, not in separated PL, monoacylglycerol (MAG), diacylglycerol (DAG) and free fatty acid (FFA) fractions (Supplementary Fig. 2B, D–F). Across all diets, the level of total male pup liver PAM peaked at P10 and significantly decreased at P21, and day 35 (relative percentage: $p < 0.0001$, concentration: $p = 0.0024$).

We observed female pup brain and liver tissue total PAM levels were similar to that of male pups fed the LP, MP, and HP diets from birth (Supplementary Fig. 3A, B).

Similar to male pup tissue levels of PAM, there was not a significant main effect of diet, only a significant main effect of time ($p < 0.0001$ for all) on the concentration of other male pup lipogenic fatty acid acids in the brain including palmitoleic acid (16:1n-7; POA) and stearic acid (STA; 18:0) (Supplementary Fig. 4A, B). In contrast, there was a significant diet x time interaction on the concentration ($p = 0.0036$) and relative percentage ($p < 0.0001$) of liver POA (Supplementary Fig. 5A), in which male pups fed the LP diet had significantly lower concentration of liver POA than male pups fed the MP and HP diet at P0, but not P10-35. However, male pups fed the LP diet also had a significantly lower relative percentage of liver POA ($p < 0.0001$) than mice fed the HP diet at all timepoints examined (Supplementary Fig. 5A). Additionally, there was a significant main effect of diet on liver oleic acid (18:1n-9; OLA) whereby male pups fed the LP diet (containing high OLA) had a higher concentration ($p = 0.0005$) and relative percentage ($p < 0.0001$) of liver OLA than male pups fed the MP and HP diet (containing equal parts of dietary OLA to PAM, and low dietary OLA, respectively) (Supplementary Fig. 5C). Lastly, there was not a significant effect of diet, only a significant main effect of time ($p < 0.0001$ for all) on the concentration of other n-6 polyunsaturated fatty acids (PUFA) including linoleic acid (LNA; 18:2n-6) and arachidonic acid (ARA; 20:4n-6) in the male pup brain (Supplementary Fig. 4D, E) and liver (Supplementary Fig. 5D, E).

## Pup carbon isotope ratio analysis

To investigate the dietary origin of sources maintaining the pup brain pool, we measured male pup tissue $\delta^{13}$C-PAM. There was a significant diet by time interaction on male pup brain $\delta^{13}$C-PAM ($p = 0.0008$) (Fig. 1C). Across all timepoints, there was a significant dose-response increase to lower levels of dietary PAM in male pup brain $\delta^{13}$C-PAM, whereby lower dietary PAM resulted in a more enriched brain

$\delta^{13}$C-PAM value, indicative of an increased contribution of DNL from dietary sugars. Differences in male pup brain $\delta^{13}$C-PAM were largest between the LP and HP diet groups, in which male pup brain $\delta^{13}$C-PAM was 4–11% significantly more enriched at all timepoints in mice fed the LP diet, compared to the HP diet ($p = 0.0476$ to $< 0.0001$ for all adjusted $p$ values; Supplementary Table 1). Male pup brain $\delta^{13}$C-PAM was also 4–7% significantly more enriched in pups fed the LP compared to the MP diet at P0, P21 and day 35 ($p = 0.0085$ to $< 0.0001$ for all adjusted $p$ values; Supplementary Table 1). Differences in male pup brain $\delta^{13}$C-PAM between the MP and HP diet were only observed at day 35, in which male pup brain $\delta^{13}$C-PAM was 4% significantly more enriched in male pups fed the MP diet, compared to the HP diet ($p = 0.0066$; Supplementary Table 1). Male pup brain $\delta^{13}$C-PAM was also less enriched at earlier vs. later postnatal timepoints in male pups fed the LP diet. However, in male pups fed the HP diet, brain $\delta^{13}$C-PAM was more enriched at earlier vs later timepoints. For visual purposes, only male pup brain $\delta^{13}$C-PAM means significantly different by diet at each timepoint are displayed in Fig. 1C; for all diet by time male pup brain $\delta^{13}$C-PAM interaction effects by Tukey's multiple comparisons test refer to Supplementary Table 1.

To investigate the dietary origin of sources maintaining the pup liver pool, we measured male pup liver $\delta^{13}$C-PAM. There was a significant main effect of diet ($p < 0.0001$) and time ($p < 0.0001$) on male pup liver $\delta^{13}$C-PAM (Fig. 1D). Across all timepoints, there was a significant dose-response increase in male pup liver $\delta^{13}$C-PAM in response to lower levels of dietary PAM similar to the brain, that was more pronounced. Differences in liver $\delta^{13}$C-PAM between the LP and HP diet groups were the largest in which liver $\delta^{13}$C-PAM was 18–31% significantly more enriched at all timepoints in male pups fed the LP compared to HP diet ($p < 0.0001$). Male pup liver $\delta^{13}$C-PAM was also 13–24% significantly more enriched at all timepoints in male pups fed the LP diet, compared to the MP diet ($p <0.0001$). The significant dose-response increase in total liver $\delta^{13}$C-PAM in response to lower levels of dietary PAM in male pups persisted in a subset of separated male pup liver TAG ($p = 0.0002$) (Supplementary Fig. 6A), however, not in the separated liver PL, CE, MAG, DAG, and FFA fractions at P35 (Supplementary Fig. 6B–F).

We observed female pup tissue $\delta^{13}$C-PAM was similar to that of male pups fed the LP, MP, and HP diets from birth (Supplementary Fig. 3C, D).

Similar to male pup brain $\delta^{13}$C-PAM, there was a significant diet x time interaction on other male pup lipogenic brain and liver fatty acid $\delta^{13}$C values, including brain $\delta^{13}$C-STA ($p = 0.0036$) (Supplementary Fig. 7B) and liver $\delta^{13}$C-STA ($p = 0.0076$) (Supplementary Fig. 8B), as well as brain $\delta^{13}$C-OLA ($p = 0.0398$) (Supplementary Fig. 7C). Additionally, there was a significant main effect of diet on brain ($p < 0.0001$) and liver ($p < 0.0001$) $\delta^{13}$C-POA (Supplementary Figs. 7A and 8A, respectively), whereby male pups fed the LP diet had more enriched $\delta^{13}$C-POA, compared to male pups fed the MP and HP diets in both tissues. Although male pup liver $\delta^{13}$C-STA depended on time ($p = 0.0076$), similar to male pup liver $\delta^{13}$C-PAM, and liver $\delta^{13}$C-POA, liver $\delta^{13}$C-STA also followed a dose-response increase in $\delta^{13}$C in response to low dietary PAM (Supplementary Fig. 8B). In contrast, although male pup liver $\delta^{13}$C-OLA also depended on time ($p = 0.0238$), liver $\delta^{13}$C-OLA appears to be the reverse; a dose-response increase in $\delta^{13}$C-OLA in response to high dietary PAM (and low dietary OLA) (Supplementary Fig. 8C). Like male pup tissue $\delta^{13}$C-PAM, any differences between diet groups were more pronounced overall at the level of the liver, compared to the brain for $\delta^{13}$C-POA, -STA and -OLA (Supplementary Fig. 7A, B, C vs. Supplementary Fig. 8A, B, C). Interestingly, compared to male pup brain $\delta^{13}$C-PAM and $\delta^{13}$C-POA which displayed a similar dose response increase in $\delta^{13}$C in response to low dietary PAM as liver $\delta^{13}$C-PAM and -POA, differences in male pup brain $\delta^{13}$C-STA and -OLA were less clear (Supplementary Fig. 7B, C) and did not follow a similar response as liver $\delta^{13}$C-STA and -OLA (Supplementary Fig. 8B, C).

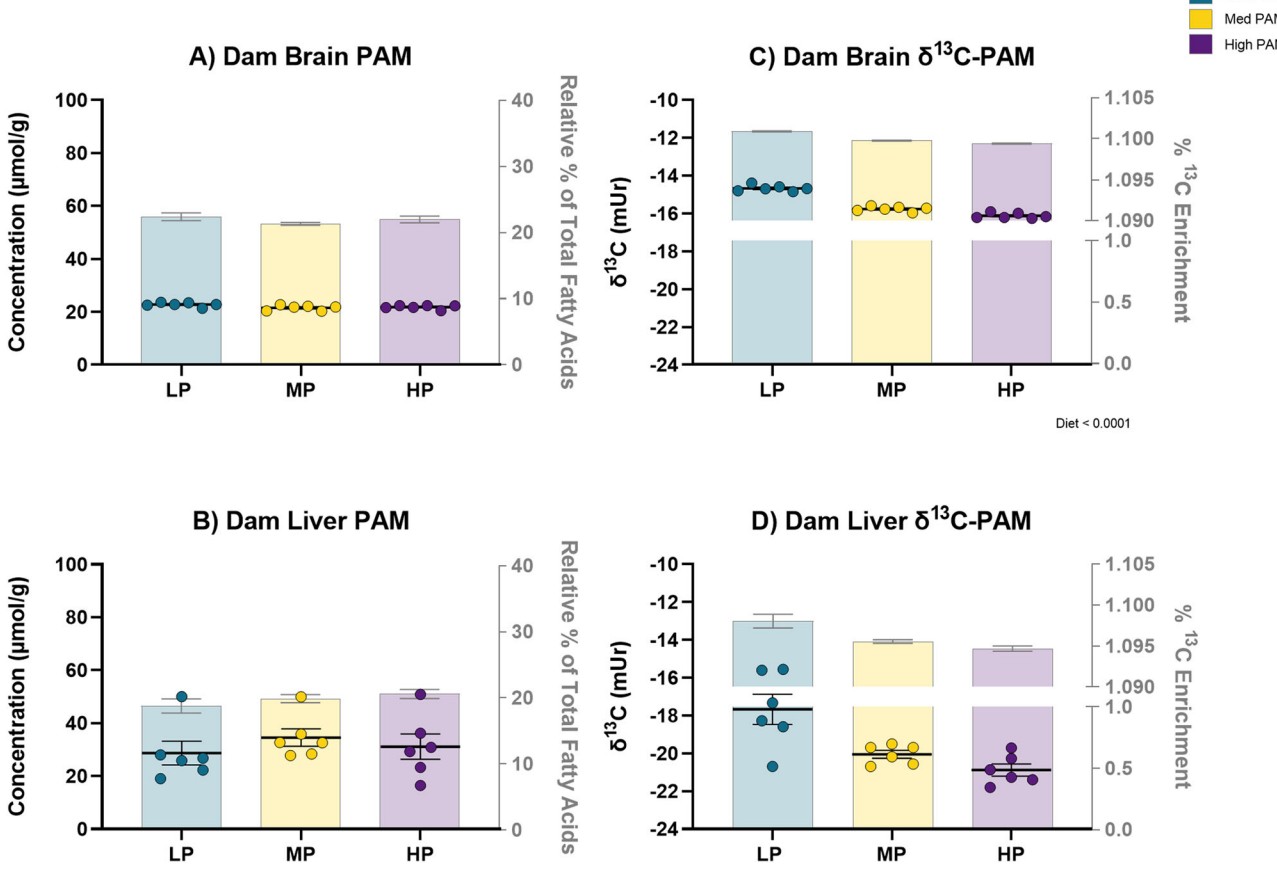

**Fig. 2 | Dam tissue palmitic acid levels are maintained primarily by lipogenesis from dietary sugars augmented in dams fed low palmitic acid.** The levels of dam brain palmitic acid (PAM) (**A**) and liver PAM (**B**) were not significantly affected by diet. However, there was a significant effect of diet in both dam brain $\delta^{13}$C-PAM (**C**) ($p < 0.0001$) as well as dam liver $\delta^{13}$C-PAM (**D**) ($p = 0.0013$). Data are biological replicates presented as mean ± SEM analyzed by ordinary one-way analysis of variance (ANOVA) (**A**, **C**, **D**) or Kruskal–Wallis test (**B**); $n = 6$ dams per diet.

Individual data points represent concentration data (for which figure statistics correspond to) while the bars show relative percentage data (**A**, **B**). Data points represent $\delta^{13}$C-PAM values (for which figure statistics correspond to) while the bars show %$^{13}$C enrichment ($^{13}$C/($^{13}$C+$^{12}$C)) (**C**, **D**). Source data are provided as a Source Data file. $\delta^{13}$C, $^{13}$C/$^{12}$C; HP high palmitic acid, LP low palmitic acid, Med medium, MP medium palmitic acid, PAM palmitic acid.

Importantly, male pup $\delta^{13}$C values of other n-6 PUFAs LNA and ARA, as well as n-3 PUFA docosahexaenoic acid (22:6n-3; DHA) were not affected by diet in either the brain (Supplementary Fig. 7D–F) or the liver (Supplementary Fig. 8D–F).

**Pup sensorimotor development tests**
There were no differences in average pup litter scores for both the geotaxis test (Supplementary Fig. 9A) and the righting reflex test (Supplementary Fig. 9B) between the three diet groups from P0-P10.

**Dam tissue PAM levels**
Liver and brain tissues were collected from dams that consumed the LP, MP, or HP diet from the start of the study until P21 when weaning commenced, a total of 71 days, to assess the effect of diet on dam tissue PAM levels. There was no significant effect of diet on the relative percentage or concentration of dam brain PAM levels (Fig. 2A) or liver PAM levels (Fig. 2B).

**Dam carbon isotope ratio analysis**
To investigate the dietary origin of sources maintaining the dam brain and liver PAM pool, we measured dam tissue $\delta^{13}$C-PAM. Similar to male pup tissue $\delta^{13}$C-PAM, there was a significant dose-response increase to lower levels of dietary PAM in both dam brain ($p < 0.0001$) and liver ($p = 0.0013$) $\delta^{13}$C-PAM (Fig. 2C, D).

**Dam maternal nest tests**
Dam nests were scored between gestational days 15–18 if they exhibited a central hollow (Supplementary Fig. 9C). There were no differences in maternal nesting behaviour between the three diet groups (Supplementary Fig. 9D).

**Day 35 RNASeq quality control**
Principal component analysis (PCA) was used to assess overall similarity between samples and revealed distinct clusters between male day 35 brain and liver tissue using the first two components, with the liver having more variance in expression than the brain (Fig. 3A). Although distinct clusters were not observed by diet for the brain (Fig. 3B), or liver (Fig. 3C), it was verified that all samples within a diet group for each tissue had an intra-group coefficient value > 0.9 before proceeding with analyses.

**Day 35 differential gene analysis**
Liver differential gene analysis revealed many differentially expressed genes (DEGs) between the LP and MP diet, and LP and HP diet in day 35 males (adjusted $p$ value < 0.05 or <0.1, log2 fold change > 1.2) (Supplementary Table 2), however, no DEGs were detected between the MP and HP diet. Utilizing the same cut-off criteria, we did not identify any DEGs between any diet group comparisons in the brain. Therefore, volcano plots were only constructed to explore liver DEGs in detail (Fig. 4). The protein-coding DEG with the highest log2 fold

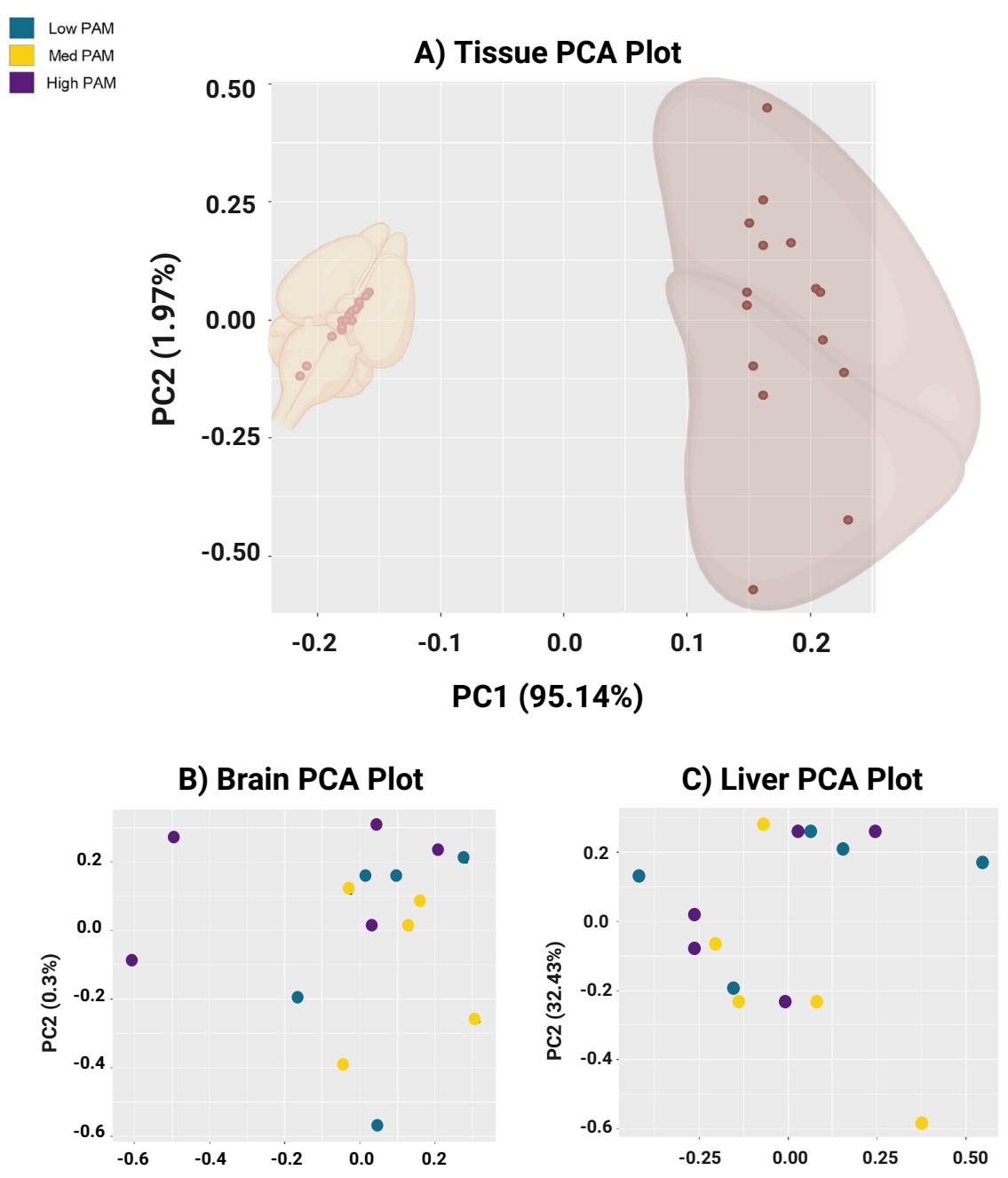

**Fig. 3 | Male day 35 mRNA expression is more variable in the liver than the brain.** Although principal component analysis (PCA) did not reveal distinct clusters by diet group in the male brain (**B**) or liver (**C**), there were distinct clusters by PCA between brain and liver tissue (**A**), in which the liver displayed more variance in expression than the brain. Data points are individual sequenced samples; *n* = 15 male mice per tissue (biological replicates) (**A**), *n* = 5 male mice per diet per tissue (biological replicates) (**B**, **C**). Med medium, PAM palmitic acid, PC1 principal component 1, PC2 principal component 2. Created with BioRender.com.

change in the male liver between the HP and LP diet group was *cyp7a1* (log2 fold change 3.97; *p* value 5.9E−10); encoding for the enzyme cholesterol 7 alpha-hydroxylase which was upregulated in mice fed the HP diet, compared to the LP diet (Fig. 4A). The protein-coding DEG with the highest log2 fold change in the liver between the MP and LP diet group was *pla2g4f* (log2 fold change 3.66; *p* value 2.88E−05) encoding for calcium-dependent phospholipase A$_2$ which was upregulated in mice fed the MP diet, compared to the LP diet (Fig. 4B).

**Day 35 pathway analysis**

To explore differences in genes at a pathway level between diet groups at day 35 in male mice, gene set enrichment analysis (GSEA) was conducted using a soft threshold of 6 and minimum module size of 20. Contrasting the HP and LP diet groups, the top 30 significant pathways identified for the male day 35 liver are presented in Fig. 5 and Supplementary Table 3. The differential expression of *cyp7a1* persisted in the pathway analysis, where *cyp7a1* was identified in 10 of the significant gene pathways contrasting the HP and LP diet for the liver. Importantly, multiple pathways containing genes central to lipid metabolism were identified in pathways contrasting the HP and LP diet groups; namely the upregulation of import across the plasma membrane and biosynthesis of cholesterol, and the downregulation of acyl-CoA and thioester synthesis (FDR = 0.2) (Fig. 5). Brain GSEA contrasting the HP and LP diet groups in day 35 males resulted in insignificant

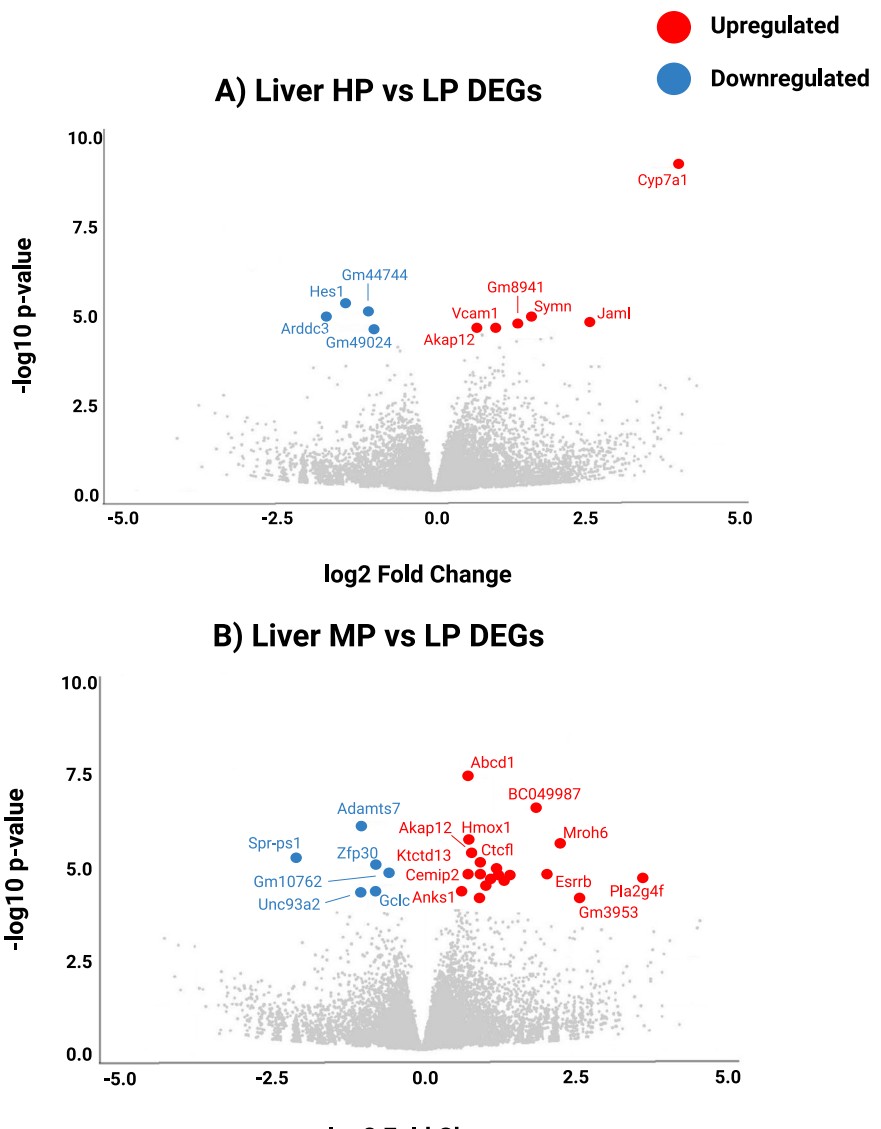

**Fig. 4 | Differentially expressed genes identified in the male liver but not the brain at day 35.** Each red and blue point represents up- and down-regulated differentially expressed genes (DEGs), respectively, identified in day 35 male liver but not the brain between male mice fed the high PAM compared to low PAM diet (**A**) and the medium PAM compared to low PAM diet (**B**). Data points are individual DEGs. The Wald test was used to identify DEGs between samples in R Studio, with an adjusted *p*-value of 0.05, and log2 fold change was set to 0.58 for visualization purposes in volcano plot construction. For all DEGs with log2 fold change > 1.2 refer to Supplementary Table 2. *n* = 5 male mice per diet (biological replicates). DEG differentially expressed gene, HP high palmitic acid, LP low palmitic acid, MP medium palmitic acid. Created with BioRender.com.

changes (Supplementary Table 3). Contrasting the MP and LP diet groups, the top 30 significant pathways for the day 35 male liver and brain are presented in Supplementary Table 4, however, only differentially expressed pathways in the liver were statistically significant (FDR = 0.2). The differential expression of *pla2g4f* did not persist in pathway analysis contrasting the MP-LP diet GSEA for the male day 35 liver as pathways identified were related mostly to the cell cycle; and unique to the HP to LP dietary contrast for the liver, no pathways were identified pertaining to lipid metabolism between the MP and LP diet groups (FDR = 0.2) (Supplementary Table 4). Because no DEGs were identified between the HP and MP diet groups for the male day 35 liver or brain, GSEA was not conducted contrasting the HP and MP diet.

**Day 35 network analysis**
To explore modules of highly correlated genes, weighted gene co-expression network analysis (WGCNA) was conducted on the top 1000 most variable genes using normalized count data from each tissue at

day 35 in male mice (Supplementary Fig. 10). Gene Ontology (GO) enrichment analysis was performed on all identified modules. For the male day 35 liver, a network of 988 genes was divided into 9 modules (Fig. 6A). Enrichment analysis revealed the yellow module (138 genes) to be significantly enriched in biological processes related to lipid metabolism including lipid, sterol, and small molecule biosynthesis, as well as acyl-CoA, thioester, and fatty acid metabolic processes (Fig. 6B). The top 10 genes for the yellow module included those involved in fatty acid and cholesterol metabolism and synthesis (*acot2*, *sqle*, *cyp51*), fatty acid elongation (*elovl 6*) as well monounsaturated fatty acid synthesis (*scd2*), and genes involved in DNL (*fasn, acly*) (Fig. 6C). The blue module (185 genes) was also enriched in biological processes related to lipid metabolism including lipid, unsaturated fatty acid, icosanoid, arachidonic acid, long-chain fatty acid, small molecule, fatty acid, and monocarboxylic acid metabolic processes (Supplementary Table 5). The DEG *pla2g4f* was identified among total genes in the blue module, and other members of the cytochrome P450 family

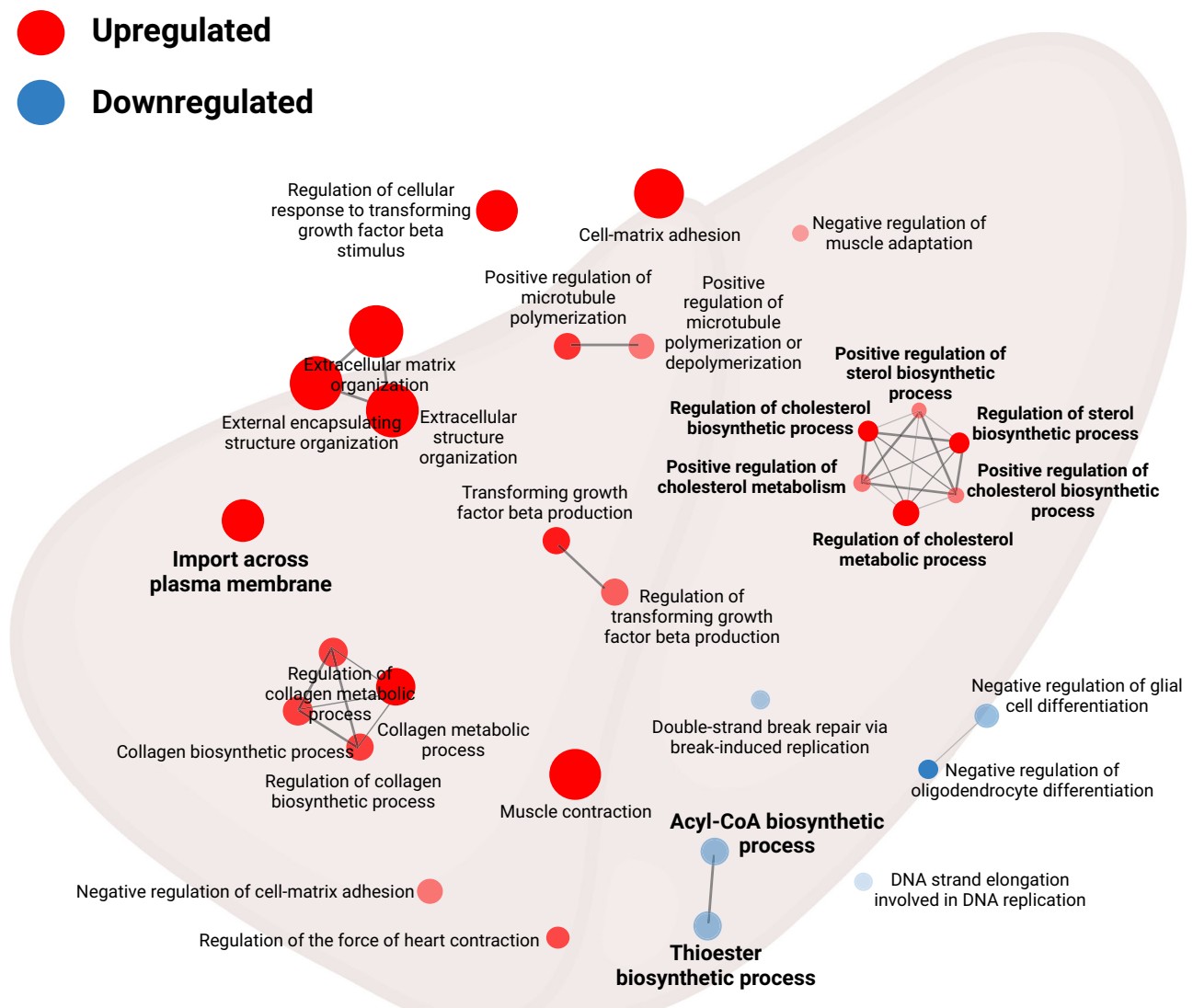

**Fig. 5 | Differential expression of pathways involved in lipid metabolism identified in the male liver, but not the brain between mice fed high palmitic acid compared to low palmitic acid at day 35.** Contrasting male mice fed the high palmitic acid (PAM) diet compared to the low PAM diet at day 35, genes central to lipid metabolism in pathways including import across the plasma membrane and cholesterol synthesis were upregulated (false discovery rate (FDR = 0.2)), while acyl-CoA and thioester synthesis were downregulated (FDR = 0.2) by gene set enrichment analysis. Red and blue points represent up- and down-regulated gene pathways, respectively. $n$ = 5 male mice per diet (biological replicates). PAM palmitic acid. Created with BioRender.com.

(*cyp2c67*, *cyp7b1*) and carboxylesterases (*ces3b*, *ces3a*) were identified among the top 30 genes in the blue module. The DEG *cyp7a1* was identified in the green module (90 genes), which was enriched for homeostatic processes (Supplementary Table 5). The rest of the modules, including the turquoise, brown, red, black, pink, and magenta modules were significantly enriched in biological processes related to the cell cycle, cytokine production/inflammatory response, apoptotic processes, ATP metabolic processes, proteolysis/digestion, and supramolecular fiber organization, respectively.

For the brain, a network of 743 genes were divided into 8 modules. No significant enrichment in biological processes were found in 2 of the 8 modules identified (turquoise, yellow) (Supplementary Fig. 11) and the remaining 6 modules were not significantly enriched for any pathways related to lipid metabolism (Supplementary Fig. 11).

## Discussion

The current study utilized naturally occurring carbon isotope ratios to uncover the origin of brain PAM during development, as well as pathways involved in maintaining brain PAM at day 35 by RNA sequencing. Our study utilized isocaloric low, medium, and high PAM feeding during the pre- and post-weaning period to investigate if brain PAM is sourced from the diet, or from endogenous synthesis by DNL from dietary sugars. Although our group recently uncovered the origin of brain PAM during adulthood in mice utilizing CIRs at the natural abundance level[18,20], the current study employed brain CIRs during development to study brain PAM origin while investigating genes and pathways involved in the maintenance of brain PAM.

Despite feeding diets vastly different in levels of PAM to pups, total male pup brain PAM was maintained at all timepoints directly contrasting the level of PAM in the male pup liver which reflected dietary intake of PAM. Therefore, male pup brain $\delta^{13}$C-PAM was examined to determine the sources of PAM maintaining this pool.

The value of male pup brain $\delta^{13}$C-PAM at all timepoints was enriched overall (−14.79 to −16.47 mUr); closer to the value of dietary sugars (−11.15 ± 0.65 mUr) compared to dietary PAM (−29.44 ± 0.12 to −29.70 ± 0.19 mUr) suggesting the majority of brain PAM is maintained by DNL from dietary sugars. This was augmented in male

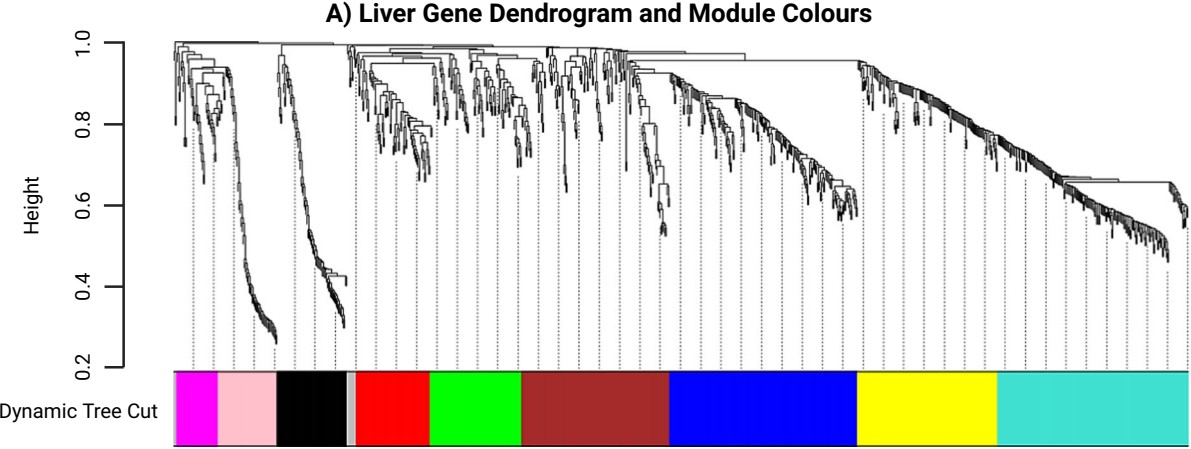

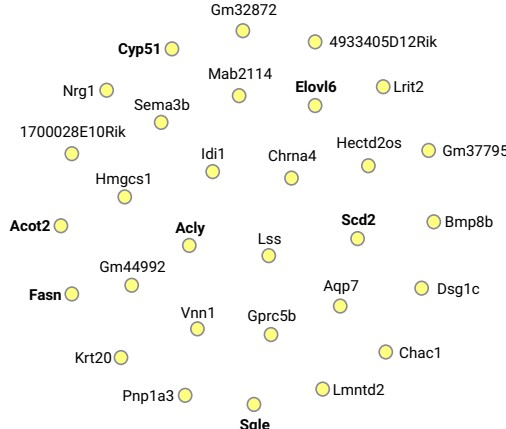

**Fig. 6 | Modules enriched for pathways and genes involved in lipid metabolism identified in the male liver but not the brain at day 35.** Of the 9 modules identified in the male day 35 liver by weighted gene co-expression network analysis (WGCNA), 3 modules were enriched for pathways and genes involved in lipid metabolism at day 35 (yellow, blue, green) (**A**). The top 10 enriched pathways in the yellow module were all related to lipid metabolism (**B**), and the top 30 genes in the yellow module contained genes central to de novo lipogenesis, fatty acid, and cholesterol synthesis, as well as fatty acid elongation and monounsaturated fatty acid synthesis (**C**). $n$ = 5 male mice per diet (biological replicates). Soft threshold was set to 6 and a minimum module size of 20 was used. GO gene ontology. Created with BioRender.com.

pups fed the LP diet, compared to the MP and HP diet, consistent with previous results from our group with low and high PAM feeding from birth to 15-weeks of age[18]. Similarly, male pup liver $\delta^{13}$C-PAM was enriched overall and augmented in male pups fed the LP diet at all timepoints examined, closer to the signature of dietary sugars compared to dietary PAM. Our result suggests DNL from dietary sugars maintains the liver PAM pool as well. Enrichment in both male pup brain and liver $\delta^{13}$C-PAM values raises the question of whether $\delta^{13}$C-PAM enrichment in the brain is due to local brain DNL from dietary sugars, or uptake of circulating PAM produced in the liver by DNL from dietary sugars and subsequently incorporated into the brain. Previously, our group utilized mixed-modelling to correct for the major pool supplying the brain - blood $\delta^{13}$C-PAM - and found DNL from dietary sugars accounts for upwards of >70% of the total brain PAM pool, and 44-58% was due to local brain DNL in response to both standard and varying levels of dietary PAM in adult male and female mice[18,20]. However, previous mixed modeling does not resolve which tissue is upregulating DNL from dietary sugars in response to diet. Therefore, RNAseq was conducted to identify genes and pathways that may be involved in maintaining total male brain PAM at day 35, as this was the timepoint mice consumed the diets the longest both pre- and post-weaning.

Compared to the liver, the male pup brain had a relatively unchanged mRNA expression profile at day 35. Furthermore, pathway analysis revealed significant differences in affected pathways in the liver between HP-LP diet comparisons and MP-LP diet comparisons, whereas no pathways were significantly differentially expressed in the male brain between all dietary group comparisons. Importantly, pathways involved in lipid metabolism were unique to the HP-LP diet contrast, in which acyl-CoA and thioester biosynthetic processes were downregulated in mice fed the HP diet compared to the LP diet at day 35 which both included genes involved in converting pyruvate to acetyl-CoA (*dlat, pdk1, pdhx, pdk4, pdhb, mpc1, mpc2, pdha1, acat1, pdk3, pdk2*), converting long- (*acsl1, acsl5, acsl4*) and short- chain (*acss1, acss2*) fatty acids to acyl-CoA products, DNL (*acaca, acly, mlycd, acacb*) and the synthesis of long-chain saturated and mono-unsaturated fatty acids (*elovl 7, elovl 1, elovl 5, elovl 3, elovl 6*). This was consistent with network analysis, in which 3 of the 9 modules were identified to be related to lipid metabolism in the liver and differentially expressed by diet, while no modules were identified to be related to lipid metabolism in the brain. The yellow module in Fig. 6 was most enriched for lipid metabolic processes pathways, including acyl-CoA and thioester metabolic processes pathways that were identified in pathway analysis. Top genes in the yellow module included those

involved in DNL (*fasn*, *acly*) and the synthesis of very long-chain saturated and monounsaturated fatty acids (*elovl 6*, *scd2*). Altogether, the RNAseq data suggests that the male liver is upregulating DNL from dietary sugars to supply the brain with PAM at day 35.

Previous work on the origin of brain PAM during development reported that intact perdeuterated and trideuterated PAM fed to artificially reared rat pups between 6 and 14 days old did not enter brain tissue lipids; concluding the brain relies on local endogenous synthesis to maintain brain PAM levels[9,10]. While our present study and previous work supports the notion that the brain heavily relies on endogenous synthesis to maintain PAM levels compared to dietary PAM overall during development, we also show that genes involved in hepatic DNL from dietary sugars are upregulated compared to the brain in response to low PAM at day 35, notwithstanding the fact the brain has a basal level of DNL[18,20]. Although we only have genetic data at day 35 in support of this finding, male pup brain, and liver $\delta^{13}$C-PAM for P0, P10 and P21 followed a similar dose-response increase in tissue $\delta^{13}$C-PAM in response to low dietary PAM levels. Furthermore, dose-response increases in liver $\delta^{13}$C-PAM to low dietary PAM levels were more pronounced from P0-P21, compared to day 35. Therefore, we expect hepatic DNL from dietary sugars is also upregulated to supply the male pup brain from P0-P21, perhaps to an even greater extent. Nevertheless, a limitation of our work includes the fact lipogenic genes are regulated by complex mechanisms, for example, by transcription factors sterol regulatory element binding protein-1c and carbohydrate element-binding protein - which are responsive to cell signaling and intermediates of glycolysis[22–24]. Therefore, in addition to gene expression, future studies should measure post-transcriptional, post-translational, and allosteric regulators of lipogenesis to gain a more holistic understanding of lipogenesis in response to diet.

Importantly, other molecules including fatty acids, ketone bodies, and ketogenic amino acids can contribute carbon to the acetyl-CoA pool used for hepatic DNL[25] in addition to the primary substrate of glucose in rats and mice[19]. The contribution of other molecules used in the synthesis of PAM therefore results in the depletion of tissue $\delta^{13}$C-PAM. In the brain, it has been suggested ketones are the main substrate for brain cholesterol and fatty acid synthesis particularly during development[26–29]. Although brain $\delta^{13}$C-PAM in our study was enriched overall suggesting the majority of male brain PAM is maintained by DNL from dietary sugars, male brain $\delta^{13}$C-PAM irrespective of diet at all timepoints had some degree of depletion (−14.79 to −16.47 mUr) from the dietary sugars (−11.15 ± 0.65 mUr) suggesting a contribution of other depleted molecules supplying acetyl-CoA for DNL in addition to dietary sugars. Furthermore, we observed male pup brain $\delta^{13}$C-PAM depended on time in which male pups fed the LP diet had more enriched $\delta^{13}$C-PAM later, compared to earlier during development while male pups fed the HP diet had the reversed observation. This suggests male pups fed the LP diet may require the use of other substrates in addition to dietary sugars earlier in development to produce PAM for the brain due to the low supply of preformed PAM to keep up with brain energy demands as compared to later in development. However, since the brain has a basal level of DNL (44-58%)[18,20] we would anticipate a more depleted male pup brain $\delta^{13}$C-PAM value than a liver $\delta^{13}$C-PAM value overall if it were the case that ketones were the primary substrate for DNL in the brain and we observed the opposite. Future studies should characterize the contribution of other molecules that contribute to tissue acetyl-CoA pools used in synthesizing PAM during development in addition to dietary sugars, as a limitation of our work is the inability to characterize and quantify other molecules which contribute to the depletion of brain $\delta^{13}$C-PAM, other than dietary fatty acids and protein measured here. Furthermore, future studies could measure hydrogen isotopes of water incorporated into PAM sourced from lipogenesis, which enables quantification of the fraction of newly synthesized PAM. Nevertheless, on a broader scale, our technique could be applied in studying developmental or degenerative

disorders to better understand the relationship between diet and the developing or aging brain. For instance, measuring $\delta^{13}$C values of cholesterol precursors in the study of inborn errors of cholesterol biosynthesis, or, measuring $\delta^{13}$C values of brain fatty acids to study lower glucose uptake, glucose hypometabolism and glucose hypermetabolism in the case of diabetes, Alzheimer's disease, and cancer, respectively.

Interestingly, male pup brain $\delta^{13}$C-PAM values observed here are consistent, and slightly more enriched than $\delta^{13}$C enrichment found in human brain $\delta^{13}$C-PAM values (average -19.95 ± 1.17 mUr)[3], the latter likely a product of controlled animal feeding compared to a variable human diet. Data on individual fatty acids and their effect on DNL is limited, in addition to studies comparing the lipogenic potential of the liver compared to the brain in response to alterations in dietary fatty acids, making our study unique. Nevertheless, our study is consistent with others in the field that have used diets depleted in the plant-based precursor alpha-linolenic acid (18:3n-3; ALA) for DHA synthesis and demonstrated increased synthesis of DHA from ALA in the liver[30], but not the brain[31]. More specifically, dietary deprivation of n-3 PUFA (0.2% ALA, no DHA) in post-weaning rats for 15 weeks infused with $^{14}$C-ALA results in increased synthesis of DHA from ALA 6.6-, 8.4-, and 2.3-fold in liver TAG, PL, and CE fractions, respectively[30], but not in the brain[31]. Importantly, the expression of enzymes governing the conversion of ALA to DHA (*elovl-2*, *elovl-5*, $\Delta 5$-$\Delta 6$-*desaturase*) were found to be upregulated in liver, but not the brain of rats fed the n-3 PUFA deficient diet[32]. These studies are consistent with the concept observed here that compared to the brain, the liver is metabolically flexible in synthesizing fatty acids in response to changes in dietary supply.

In addition to the liver, enzymes involved in DNL (*acaca*, *scd*, *fasn*, etc.) have been identified in the mammary tissue at different lactation stages in the mouse[33–35] in order to synthesize fatty acids to produce a large amount of milk TAG in addition to obtaining fatty acids from the blood[36]. It is therefore unsurprising that the dams in our study fed the LP diet were able to produce milk with ample amounts of PAM assessed through the pup stomach content at P0 and P10, a limitation of our work (Supplementary Fig. 12). Nevertheless, dam milk fed to the pups during the pre-weaning period still contained low (21-28%), medium (25-35%), and high (31-41%) levels of PAM and pups consumed the diets post-weaning for 14 days, enabling us to investigate low, medium, and high PAM feeding both pre- and post-weaning irrespective of this adaptation. The reflection of dietary PAM in the level of dam milk PAM observed here is consistent with fat-free dam feeding resulting in the production of PAM in dam milk (27.7 ± 1.3 moles/100g lipid)[37]. Importantly, the study by S. Smith et al. 1969 also found the lipogenic capacity of the liver increased during lactation compared to non-bred mice, and the liver, but not the mammary gland had altered activity of enzymes involved in DNL[37]. Nevertheless, 23.6% of PAM in dams fed the fat-free diet was synthesized from dietary sugars by DNL in the mammary gland in the study by Smith et al.[37] suggesting that a basal level of DNL occurs in the mammary gland. Importantly, total PAM in the dam liver is unaffected by diet in the present study, likely a result of long-term diet consumption, yet, dam liver $\delta^{13}$C-PAM is significantly higher in dams fed the LP, compared to the MP and HP diets. Therefore, PAM levels in dam milk observed in the present study may reflect upregulated hepatic DNL in addition to a basal level of DNL in the mammary gland.

Regardless of hepatic lipogenic compensation, the brain invests in a basal level of DNL to synthesize PAM[18,20] that we now show, based on RNAseq data, is not responsive to levels of PAM in the diet compared to the liver at day 35. This is similar to what is observed in cancer cells in which *fasn* is over-expressed compared to healthy cells, and non-responsive to nutritional cues[38–40]. Interestingly, it was demonstrated in breast cancer cells that the elevated basal level of DNL does not have a quantitative nor qualitative function[41]. Contrarily, in non-cancerous

models DNL has been suggested to preserve physical performance in muscle in response to disrupted metabolic homeostasis[42] and to serve as a crucial source of fatty acids comprising the membrane of autophagosomes for autophagy in the healthy adipocyte[43]. In the brain, *fasn* depletion in the neural retina and Schwann cells results in neurodegeneration and blindness[44] as well as defects in myelination and impaired PPAR-γ regulated transcription[45], respectively. Alternatively, the synthesis of PAM requires NAD$^+$ to serve as an electron acceptor (14 NADPH), therefore, it is also possible the brain invests in DNL to provide NAD$^+$ to support the overall metabolic demand of the brain[46], which may be a different environment than the adult brain where glucose is selected as the major source of energy compared to lipid[47]. It has been demonstrated under hypoxic conditions neuronal cells metabolize glutamate and glutamine to saturates like PAM in culture[48] in which oxidized NAD$^+$ supports anaerobic glycolysis[49]. Furthermore, it was recently demonstrated that there is a crucial role of NAD$^+$ availability for DNL in proliferating cancer cells[50]. Future studies should explore if the utilization of dietary sugars for DNL in non-cancerous, non-hypoxic conditions provides NAD$^+$ to support overall brain energy requirements during development.

Overall, our study shows the upregulation of genes involved in hepatic DNL from dietary sugar supplies the male murine brain with PAM at day 35, augmented in response to low levels of dietary PAM fed from birth. We also show brain CIRs of PAM are responsive to levels of dietary PAM during the pre- and post-weaning period, putting forward a feasible measure of tissue PAM origin and homeostasis during development, under the condition dietary precursors and products are well-defined. Importantly, in response to low dietary PAM, we show that there is compensation in both the dam and the pup to ensure total brain PAM pools are maintained by the upregulation of fatty acid synthesis by the liver, suggesting the importance of PAM regulation during development.

## Methods

### Animals
The study was not pre-registered on any repository. The animal use protocol (20012290), conducted in accordance with the Canadian Council on Animal care, was approved by the animal ethics committee at the University of Toronto. Thirty-two 4-week-old BALB/c dams were acquired from Charles River Laboratories (Saint Constant, QC, CA; RRID: IMSR_CRL:028) and housed within the Division of Comparative Medicine at the University of Toronto which maintained a constant light cycle (12 h light/12 h dark), controlled temperature (21 °C), and 40–60% humidity throughout the study in which food and water were available ad libitum. Using simple randomization procedures with stratification for body weight, dams were equilibrated onto either a LP, MP, or HP diet for 4 weeks and grouped-housed (4 females per cage), after which they were bred in-harem (1 male:2 females per cage) with 8-week-old BALB/c males (RRID: IMSR_CRL:028) to produce one generation of offspring.

Of the 32 bred dams, 29 dams (91%) had litters at the end of gestation, however, 2 dams and their litters were excluded due to the occurrence of dam infanticide (Supplementary Table 6) to reduce between-litter variance. Importantly, there were no significant differences between diet groups in the total number of pups, nor the total number of female or male pups born per dam (Supplementary Fig. 13). Nevertheless, although the ratio of male:female pups born was not different by diet, all dams produced a significantly higher number of male pups, compared to female pups ($p < 0.0001$) (Supplementary Fig. 13D) in our study. Therefore, our primary study outcomes ($\delta^{13}$C-PAM and RNAseq) were powered to assess male pups, and female pups were included, when possible, to supplement male data. Same-sex pup siblings were not pooled for outcomes and the distribution of litter representation across outcomes is reported in Supplementary Table 6.

Offspring maintained the respective dam diet and were housed with their littermates and dam until weaning at P21, after which they were group-housed (3–4 per cage) by sex until P35. Male, and when possible female pups at P0 ($n = 5$–8 males, 2-3 females per diet) and P10 ($n = 4$–7 males, 2–3 females per diet) were humanely euthanized by cervical dislocation while male, and when possible female pups at P21 ($n = 6$–7 males, 1–3 females per diet) and P35 ($n = 5$–6 males, 1–2 females per diet), as well as 15-week-old dams at offspring P21 ($n = 6$ dams per diet) were humanely euthanized by asphyxiation followed by cervical dislocation. Brain and liver tissue were collected, snap-frozen in liquid nitrogen, and stored at −80 °C until experimentation. Pups as well as dams were handled three times weekly to minimize stress, and although 2 litters were excluded from the study due to dam infanticide, no pups met the defined exclusion criteria which included ± 10% change in weight over a short time, and/or exhibiting symptoms of a suspected illness and/or failure to thrive throughout the study.

### Diets
The three study diets were isocaloric (% by calorie: 16.8% fat, 19.4% protein and 63.9% carbohydrates), differing only in fatty acid composition and modified from the AIN-93G rodent diet. Diets were formulated at Dyets Inc. using palmitate and oleate ethyl ester (Nu Chek Prep Inc., item no. N-16-E, U-46-E). The LP (item no. 104636) and HP (item no. 104638) diets contained 15.6% energy of either OLA, or PAM, respectively, while the MP diet (item no. 104637) contained 7.8% energy from PAM, and 7.8% energy from OLA. As OLA has been previously utilized by our group as a fatty acid substitute to achieve matched total fat across different animal diets[18,51,52] and is a downstream product of PAM rather than a precursor, it was used as a substitute for PAM in the LP and MP diets. Additionally, all three study diets contained 1.2% energy from soybean oil to ensure diets were not deficient in essential PUFAs. Lipids were extracted from each diet using methods adapted from Folch et al.[53] to verify the diet fatty acid composition by gas chromatography (GC)-flame ionization detection (FID) upon arrival. The LP, MP, and HP diets contained $1.3 ± 0.4\%$, $47.3 ± 1.0\%$ and $95.6 ± 0.5\%$ PAM, respectively (Table 1). While OLA comprised $95.5 ± 2.0\%$, $49.1 ± 1.4$ % and $0.8 ± 0.9\%$ of the LP, MP, and HP diets, respectively. Dietary fatty acid $\delta^{13}$C values were confirmed by GC-combustion-isotope ratio mass spectrometry (GC-C-IRMS), while dietary carbohydrate and protein supplying greater than ~1% energy were confirmed by elemental analysis (EA-IRMS). The weighted average of all carbohydrate sources in the three diets (herein referred to as dietary sugars) was measured to be more enriched in $\delta^{13}$C (−11.15 ± 0.65 mUr), than dietary PAM (−29.44 ± 0.12 mUr to −29.70 ± 0.19 mUr), OLA (−28.26 ± 0.12 mUr to −28.39 ± 0.12 mUr) and casein (−23.88 ± 0.12 mUr) making dietary carbohydrates distinguishable from dietary fat and protein isotopically (Table 1). The importance of dietary macronutrients being distinguishable isotopically is illustrated in Supplementary Fig. 14, as it allows for the measurement of tissue $\delta^{13}$C-PAM which gives rise to fatty acid origin. Because weanling pups consumed dam milk from P0-P21, the fatty acid profile of P0 and P10 stomach content was assessed to ensure pups also consumed different levels of PAM pre-weaning. Importantly, P0 and P10 stomach content also contained low (21–28%), medium (25–35%), and high (31–41%) PAM levels (Supplementary Fig. 12) reflecting levels of dietary PAM. Feeding of the diets was performed by one investigator (MES); therefore, blinding to the study diets was not possible.

### Lipid extraction
Total lipids were extracted from all BALB/c male and female pup pulverized right brain hemispheres and liver (P0: $n = 5$–8 males, 2–3 females per diet; P10: $n = 4$–7 males, 2–3 females per diet; P21: $n = 6$–7 males, 1–3 females per diet; P35: $n = 5$–6 males, 1–2 females per diet), as well as dam brain hemispheres and liver at 17 weeks of age ($n = 6$ dams per diet) using methods adapted from Folch et al.[53] between the hours

**Table 1 | Diet composition**

| | Low PAM Diet | | Medium PAM Diet | | High PAM Diet | |
|---|---|---|---|---|---|---|
| | % Energy | δ¹³C (mUr) | % Energy | δ¹³C (mUr) | % Energy | δ¹³C (mUr) |
| **Fat (16.78%)** | | | | | | |
| Palmitic acid | - | - | 7.79% | −29.44 ± 0.12 | 15.58% | −29.70 ± 0.19 |
| Oleic acid | 15.58% | −28.39 ± 0.12 | 7.79% | −28.26 ± 0.12 | - | - |
| Soybean oil | 1.2% | - | 1.2% | - | 1.2% | - |
| **Protein (19.36%)** | | | | | | |
| Casein | 19.04% | −23.88 ± 0.12 | 19.04% | −23.88 ± 0.12 | 19.04% | −23.88 ± 0.12 |
| ʟ-Cystine | 0.32% | - | 0.32% | - | 0.32% | - |
| **Carbohydrate (63.89%)** | | | | | | |
| Sucrose | 10.64% | −11.86 ± 0.08 | 10.64% | −11.86 ± 0.08 | 10.64% | −11.86 ± 0.08 |
| Cornstarch | 38.06% | −11.01 ± 0.61 | 38.06% | −11.01 ± 0.61 | 38.06% | −11.01 ± 0.61 |
| Maltodextrin 10 | 13.34% | −10.58 ± 0.24 | 13.34% | −10.58 ± 0.24 | 13.34% | −10.58 ± 0.24 |
| | - | −11.15 ± 0.65[a] | - | −11.15 ± 0.65[a] | - | −11.15 ± 0.65[a] |

Data are presented as means ± SD, *n* = 3 technical replicates conducted in triplicate.

δ¹³C,¹³C/¹²C, PAM palmitic acid.

[a]Weighted δ¹³C average of carbohydrates.

of 09:00 and 17:00[18]. An internal standard of heptadecanoic acid (17:0) (Nu Chek Prep; item no. N-17-A) was added to all tissue homogenates at a known concentration for total fatty acid methyl ester (FAME) quantification. Total lipids were also extracted from a subset of BALB/c male pup pulverized left brain hemispheres at all timepoints (P0: *n* = 2–5; P10: *n* = 4–7; P21: *n* = 5–6; P35: *n* = 4–5 per diet), and a subset of BALB/c male pulverized liver at P35 (*n* = 5 per diet) using methods adapted from Folch et al.[53] for individual PL and neutral lipid FAME quantification, respectively, between the hours of 09:00 and 17:00[18]. Internal standards of 1–2, diheptadecanoyl-sn-glycero-3-phosphocholine (17:0-PC) (Avanti Polar Lipids Inc., Alabaster, AL, USA; item no. 850360), 1,2-diheptadecanoyl-sn-glycero-3-phosphoethanolamine (17:0-PE) (Avanti Polar Lipids Inc., Alabaster, AL, USA; item no. 830756), and 1,2-diheptadecanoyl-sn-glycero-3-phospho-ʟ-serine (sodium salt) (17:0 PS) (Avanti Polar Lipids Inc., Alabaster, AL, USA; item no. 840028) were added to left brain homogenates at a known concentration for PL FAME quantification. Internal standards of unesterified 17:0 (Nu Chek Prep; item no. N-17-A), cholesteryl heptadecanoate (17:0 CE) (Nu Chek Prep; item no. CH-816), 17:0 PC (Avanti Polar Lipids; item no. 850360) and triheptadecanoin (17:0 TAG) (Nu Chek Prep; item no. T-155) were added to liver homogenates for neutral lipid FAME quantification.

**Pup brain and liver thin layer chromatography**
Twenty-four hours prior to conducting thin-layer chromatography (TLC) to quantity PAM in separated PL fractions of male pup left brain hemispheres, silica gel H TLC-plates (Analtech, Newark, DE, USA; item no. P10011) were washed in 2:1 volume of chloroform (Sigma; item no. CX1055) methanol (Sigma; item no. 179337). H TLC-plates were activated at 100°C for 1 hour. Samples and authentic reference standards were loaded onto 1.5 and 1 cm lanes, respectively, after which PL fraction separation was achieved by incubating the plate for ~2.5 h in 30:9:25:6:18 chloroform (Sigma; item no. CX1055): methanol (Sigma; item no. 179337): 2-propanol (Sigma; item no. I9516): 0.25% potassium chloride (Sigma; item no. P3911): triethylamine (Sigma; item. no T0886) (v/v/v/v/v). Bands containing PL fractions corresponding to those in authentic reference standards including ChoGpl, EtnGpl, PtdIns, PtdSer and CerPCho were visualized under UV light after spraying with 0.1% 8-anilino-1-naphthalene sulfonic acid (Sigma; item no. A1028). Phospholipid bands were collected and an internal standard of unesterified 17:0 (Nu Chek Prep; item no. N-17-A) were added to PtdIns and CerPCho classes prior to methylation with 1 mL of 1% boron trifluoride (BF3) in methanol (Sigma; item no. B1252-1L) and 0.3 mL of hexane (Sigma; item no. 178918-4L) at 100 °C for 1 hour.

Twenty-four hours prior to conducting TLC to quantify PAM in separated neutral lipid fractions of male pup liver, silica gel G TLC-plates (Analtech, Newark, DE, USA; item no. P01911) were washed in 2:1 volume of chloroform (Sigma; item no. CX1055) methanol (Sigma; item no. 179337). G TLC-plates were activated at 100°C for 1 hour. Samples and authentic reference standards were loaded onto 1.5 and 1 cm lanes, respectively, after which neutral lipid class separation was achieved by incubating the plate for ~2.5 hours in 60:40:2 heptane (Fisher Chemical; item no. O3008-4): diethyl ether (Caledon; item no. 4700-1-40): acetic acid (Sigma; item no. A6283) (v/v/v). Bands containing neutral lipid classes corresponding to those in authentic reference standards including TAG, PL, CE, MAG, DAG, and FFA were visualized under UV light after spraying with 0.1% 8-anilino-1-naphthalene sulfonic acid (Sigma; item no. A1028). Neutral lipid bands were collected and an internal standard of unesterified 17:0 (Nu Chek Prep; item no. N-17-A) were added to DAG and MAG prior to methylation with 1 mL of 1% BF3 in methanol (Sigma; item no. B1252-1L) and 0.3 mL of hexane (Sigma; item no. 178918-4L) at 100 °C for 1 hour.

**Total brain and liver FAME quantification**
Using a Varian 430 GC-FID (Bruker, Billerica, MA) equipped with a SP-2560 100 m × 0.25 mm i.d. × $d_f$ 0.20 μm non-bonded poly (biscyanopropyl siloxane) capillary column (Supleco by Sigma-Aldrich, Bellefonte, PA, USA; item. no 24056), FAMEs were quantified by GC. The column oven program was initially set to 60 °C, increased at a rate of 8.5 °C/min to 170 °C and held for 6.12 min, increased at 4.3 °C/min to 175 °C, increased at 1.7°C/min to 185°C, increased at 0.8 °C/min to 190°C, and finally increased at 8.5 °C/min to 240°C and held for 27.76 min, totaling 66 min. An external reference standard (Nu Chek Prep; item no. GLC-569) was used to identify peaks by retention time, and subsequently, peaks were quantified by dividing the area under the curve of the peak of interest by the area under the curve of the internal standard in Compass CDS 3.0.

**Separated brain phospholipid and liver neutral lipid FAME quantification**
Using a Varian 430 GC-FID (Bruker, Billerica, MA) equipped with a DB-FFAP 30 m × 0.25 mm i.d. × 0.25 μm nitroterephthalic acid modified, polyethylene glycol, capillary column (Agilent; item no. 122-3232) FAMEs were quantified by GC. The column oven program was initially set to 50 °C for 1 min, increased at a rate of 30 °C/min to 130 °C, increased at 10 °C/min to 175 °C, increased at 5 °C/min to 230 °C and held for 9.5 min, and finally increased to 240 °C at a rate of 50 °C/min

and held for 11.13 min, totaling 40 min. An external reference standard (Nu Chek Prep; item no. GLC-569) was used to identify peaks by retention time, and subsequently, peaks were quantified by dividing the area under the curve of the peak of interest by the area under the curve of the internal standards in Compass CDS 3.0.

## Natural abundance carbon isotope ratio analysis and normalization of FAMEs by GC-C-IRMS

Male pup tissue $\delta^{13}$C-PAM, -OLA, -POA, -STA, -LNA, -ARA, and -DHA as well as dam tissue $\delta^{13}$C-PAM, and dietary $\delta^{13}$C-PAM and $\delta^{13}$C-OLA values were determined by GC-C-IRMS, carried out on a Thermo Scientific Trace 1310 GC interfaced to a MAT 253 IRMS (Thermo Finnigan MAT, Bremen Germany) via a GC Iso-link II combustion interface (Thermo Scientific). The system was equipped with a SP-2560 (100 m × 0.25 mm i.d. × $d_f$ 0.20 μm Supelco by Sigma-Aldrich; item no. 24056) capillary column. Isodat Workspace Version 3.0 (Thermo Scientific) was used to process chromatograms, in which tissue and dietary FAMEs were identified by retention time relative to external reference material (Nu Chek Prep; item no. GLC-569). Using multipoint linear normalization, IRMS tissue and dietary carbon isotopic abundance data were normalized to the international carbon isotope ratio reference scale (Vienna Peedee Belemnite; VPDB)[54–56]. Each of USGS70, USGS71, and USGS72 certified reference material (Reston Stable Isotope Laboratories, United States Geological Services, Reston, VA)[57] were injected at least once per sequence and linear regression of the measured vs. true values were used to generate normalization equations. All $R^2$ values for tissue and dietary normalization equations were >0.999. Since measurable differences in $^{13}$C-values at the natural abundance level are small, $^{13}$C-values are reported according to the universal δ notation system in order to reduce multiple leading decimals attached to $^{13}$C values at the natural abundance level[17,58,59]. Additionally, since VPDM is highly enriched (0.0112372)[56], all $\delta^{13}$C values are negative; importantly, a less negative $\delta^{13}$C value is more enriched in $^{13}$C.

## Bulk carbon isotope ratio analysis of dietary carbohydrate and protein and methylation correction by EA-IRMS

Bulk $\delta^{13}$C values of dietary carbohydrate and protein were determined by EA-IRMS at the Analytical Facility for Bioactive Molecules, The Hospital for Sick Children, Toronto, Canada, in a blinded manner. Five grams of sucrose (Dyets Inc; item no. 404400), cornstarch (Dyets Inc; item no. 401200), maltodextrin 10 (Dyets Inc; item no. 401477), and casein (Dyets Inc; item no. 400600) were obtained and stored in sealed plastic bags at room temperature until analysis, at which a known aliquot of each dietary component was transferred to smooth wall tin capsules (Elemental Microanalysis; item no. D1013) and allowed to dry overnight at ambient temperatures. Dried samples within capsules were introduced into a Flash 2000 EA (Thermo Scientific; item no. 11230245) via a closed carousel Zero Blank autosampler (Costech Analytical Technologies). Isodat Workspace Version 3.0 (Thermo Scientific) was used to process chromatograms, and bulk $\delta^{13}$C values of dietary carbohydrate and protein were normalized using the same procedure as tissue and dietary FAME $\delta^{13}$C value normalization.

Converting lipids to FAMEs involves the addition of a methyl group ($CH_3$) to the native fatty acid. Accordingly, an unmethylated 17:0 and methylated 17:0 were run by EA-IRMS in triplicate to determine the $\delta^{13}$C value of the derivatized $CH_3$. Using the correction equation below[60], the $\delta^{13}$C value of the derivatized $CH_3$ was determined to be -41.56 mUr in preparing tissue FAMEs.

Correction equation : $\delta^{13}C_{ME} = n_{FAME}\delta^{13}C_{FAME} - n_{FA}\delta^{13}C_{FA}$

All tissue $\delta^{13}$C-PAM values reported here were corrected using the determined $\delta^{13}$C value of the derivatized $CH_3$ in the following balance equation[60]:

Balance equation : $n_{FAME}\delta^{13}C_{FAME} = \delta^{13}C_{ME} + n_{FA}\delta^{13}C_{FA}$

## Statistical analysis

For differences in pups between diet groups across multiple time-points including tissue fatty acid levels, tissue $\delta^{13}$C values of fatty acids, and behaviour, two-way analysis of variance (ANOVA) was performed under the assumption of normality. Tukey's multiple comparisons test was performed in the case of a significant interaction. For differences in dams, as well as differences in pups at a single point in time, normality was assessed by Shapiro-Wilks test, and followed by either one-way ANOVA or Kruskal-Wallis's test for normally and non-normally distributed data, respectively. Mentioned statistical analyses were performed in Graph Pad Prism, Version 9 (GraphPad Software, San Diego, CA, USA). Prior to statistical analysis, all analyzed data were assessed for outliers using the ROUT method in Prism; 1 Supplementary dataset contained removed outliers ($n = 2$) using this method (Supplementary Fig. 2E, F). All significant differences were determined at $p < 0.05$.

## Illustration and figure production

Illustrations and figures were produced using BioRender.com, Graph Pad Prism Version 9, and R 4.2.1. Processing manipulations were applied to Figs. 3–6 to increase text size and quality in BioRender.com on original output from R 4.2.1 and iDEP.96.

## RNA isolation and library preparation

Total RNA was isolated from a subset of pulverized BALB/c pup left brain hemispheres at day 35 ($n = 5$ per diet) (≤100 mg starting material) and livers ($n = 5$ per diet) (≤5 mg starting material) using the RNeasy Lipid Tissue Mini Kit (Qiagen; item no. 74804) and the RNeasy Micro Kit (Qiagen; item no. 74004), respectively, according to manufacturer instructions. For liver RNA, extraction from was followed by a 3M sodium acetate (Sigma; item no. 567422) re-purification step to remove contamination. The yield and purity of RNA was initially measured photometrically using a Varioskan Lux Plate Reader (Thermo Fisher Scientific), after which, quality of total RNA samples was assessed for RNA integrity number (RIN) (average 8.27 ± 0.68) on an Agilent Bioanalyzer 2100 RNA Nano chip. The concentration of total RNA in samples was measured by Qubit RNA HS Assay on a Qubit fluorometer (ThermoFisher). After which, libraries were prepared following the NEBNext Ultra II Directional RNA Library Preparation protocol to generate stranded data. Briefly, 400 ng of total RNA was used as input material and enriched for poly-A mRNA using magnetic oligo d(T) beads, fragmented into the 200-300-bases range for 10 minutes at 94oC and converted to double stranded cDNA. Next, cDNA proceeded to library preparation and dual-index Illumina adapters were added through PCR. Of the final RNA libraries, 1 uL was loaded on a Bioanalyzer 2100 DNA High Sensitivity Chip (Agilent Technologies) to examine size. RNA libraries were quantified by qPCR using the Kapa Library Quantification Illumina/ABI Prism Kit protocol (KAPA Biosystems). Libraries were pooled in equimolar quantities and paired-end sequenced on 1 lane of S4 flowcell with the V1.5 sequencing chemistry on an Illumina NovaSeq 6000 platform following Illumina's recommended protocol to generate paired-end reads of 150-bases in length.

## Read pre-processing, alignment, obtaining gene counts

The quality of the initial sequenced data was assessed using FastQC 0.11.9. Adaptors and low-quality ends were trimmed using Trim Galore 0.5.0, and quality of trimmed reads was reassessed with FastQC. The libraries were screened for the presence of rRNA and mtRNA sequences using FastQ Screen 0.10.0. Trimmed reads were aligned to mouse reference genome using STAR aligner 2.6.0c and GENCODE release m27. RSeQC package 2.6.2 was used to assess read distribution, positional read duplication and to confirm the strandedness of alignments. Raw gene counts were obtained using HTSeq 0.6.1p2, while FPKM (fragments per kilobase per million mapped fragments) counts were generated using DESeq2 version 1.22.2 in R 4.1.1.

## RNAseq analysis

**Data pre-processing and identification of differentially expressed genes.** Overall similarity between samples was visualized using PCA and Pearson's correlation analysis in R 4.2.1. After verifying all samples sharing diet group membership presented an intra-group correlation value of $r > 0.9$, differential gene analysis was performed to determine differences in expression levels of individual genes between diet groups. Raw gene counts were used as input with DESeq2 in R 4.2.1. A subset of differentially expressed genes were identified using an adjusted $p$-value $<0.1$ or $0.05$ and log2 fold change $>1.2$ and retained for further visualization by volcano plot.

**Pathway analysis.** To determine whether pre-defined sets of genes differed between diet groups, GSEA was conducted in iDEP.96 (integrated Differential Expression and Pathway analysis; http://bioinformatics.sdstate.edu/idep/)[61,62]. Using raw gene counts as input, fold-change values returned by the DESeq2 package in iDEP were used for GSEA, independent of selected DEGs. Gene set enrichment analysis was conducted in pre-ranked mode, using an algorithm based on the fast gene set enrichment analysis (FGSEA) package[63]. The minimum and maximum gene set size was set to 5 and 2000, respectively. Using a false discovery rate of 0.2, the top 30 pathways between all diet contrasts in the liver and brain were obtained with GO Biological Process gene sets.

**Network analysis.** To determine clusters of highly correlated genes (modules) and their relationship to diet, WGCNA was conducted using the WGCNA package in iDEP. Raw counts data was normalized in iDEP by the cpm function in edgeR before WGCNA was conducted downstream on the top 1000 most variable genes. For both tissues, the soft threshold was determined to be 6 and the edge threshold was 0.4. After the co-expression network was partitioned into modules, GO enrichment analysis (GO Biological Process 2021) on each module was performed.

## Behaviour: maternal nest tests

Maternal nest tests were conducted in BALB/c dams at 12 weeks of age ($n = 5–6$ per diet) to ensure diets did not impact maternal behaviour between gestational days 15-18 using materials and a naturalistic nest score system adapted from Hess et al.[64]. Paper puck nests (Anderson's Plant Nutrient Group; item no. BRN4SR) were given to dams at 09:00 24 hours prior to scoring, after which, nests with a central hollow were photographed and measured (height of 4 corners in centimeters)[64] at 09:00 on a randomly selected day of gestation between days 15–18. Nests were then scored according to height and nest wall closure between 0 and 5, representing lower and higher quality nests, respectively[64].

## Behaviour: pup sensorimotor development tests

To ensure diets did not impact pup sensorimotor development, reflexes were evaluated between the hours of 09:00 and 12:00 by one investigator (MES) for 6 consecutive days (P0, P2, P4, P6, P8, P10) using the geotaxis and righting reflex tests with an inter-test time of 1 hour using methods adapted from other groups[65,66]. For the geotaxis test, all male and female BALB/c pups from low ($n = 8$ litters), medium ($n = 7$ litters), and high ($n = 8$ litters) PAM litters were individually placed on a 20-degree tilted surface and a 60-second cut-off time was used to measure the time it took for the pup to return their snout upwards by rotation at P0, P2, P4, P6, P8 and P10. For the righting reflex test, male and female BALB/c pups were individually placed in a restrained supine position, and a 60-second cut-off time was used to measure the time it took for the pup to return to a prone position when restraint was lifted at P0, P2, P4, P6, P8 and P10.

## Reporting summary

Further information on research design is available in the Nature Portfolio Reporting Summary linked to this article.

## Data availability

The RNA-seq data generated in this study have been reposited in the NCBI Gene Expression Omnibus (GEO) database under accession code GSE225568. The spectral data generated by GC-C-IRMS for brain $\delta^{13}$C-PAM and liver $\delta^{13}$C-PAM have been reposited in Zenodo and are accessible through https://doi.org/10.5281/zenodo.10139336 and https://doi.org/10.5281/zenodo.10140476, respectively. All other main text and supplementary data generated in this manuscript are provided in the source data file. Any additional information is available upon request to the corresponding author Dr. Richard P Bazinet (richard.bazinet@utoronto.ca). Source data are provided with this paper.

## Code availability

R code for results obtained in iDEP.96 for GSEA and WGCNA are publicly available at https://idepsite.wordpress.com/pathways/, and https://idepsite.wordpress.com/network/r, respectively. R code for PCA analysis, DEG analysis and construction of volcano plots has been reposited in Zenodo and are accessible through https://doi.org/10.5281/zenodo.10137505.

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

## Acknowledgements

The authors wish to thank Karen Ho and Bhooma Thiruv at the Centre for Applied Genomics, The Hospital for Sick Children, Toronto, Canada for their assistance with RNAseq library preparation and bioinformatics methods, respectively. Authors also wish to thank contributors at the Analytical Facility for Bioactive Molecules, The Hospital for Sick Children, Toronto, Canada, for their assistance with EA-IRMS analysis. All phases of this study were supported by a grant from the Natural Sciences and Engineering Research Council of Canada (NSERC) [RGPIN-2017-06465], R.P.B. M.E.S. was also supported by the NSERC Postgraduate Scholarship.

## Author contributions

M.E.S. contributed both design and conduct of the animal study, collection of the data, analysis of the data, and writing of the manuscript. C.T.C. and C.A.G. contributed conceptualization of the RNAseq analysis and interpretation of results with M.E.S. G.C. contributed conceptualization and conduct of the behavioural tests with M.E.S. D.K.C., K.R. and A.M. contributed conduct of thin layer chromatography with M.E.S. R.P.B. contributed conceptualization of the study design with M.E.S., supervision of all stages of data acquisition, as well as acquiring funding. All authors approve the final submission of the manuscript and agree to be accountable for the author's own contributions.

## Competing interests

R.P.B. is supported by grant funding through the Canadian Institutes of Health Research and the Natural Sciences and Engineering Research Council of Canada and holds a Canada Research Chair in Brain Lipid Metabolism. R.P.B. has received industrial grants, including those matched by the Canadian government, and/or travel support related to work on brain fatty acid uptake from Arctic Nutrition, Bunge Ltd., Capsoil Technologies, DSM, Fonterra, Mead Johnson, Natures Crops International, and Nestec Inc. Moreover, R.P.B. and C.T.C. are on the executive committee of the International Society for the Study of Fatty Acids and Lipids and held a meeting on behalf of fatty acids and cell signaling, both of which rely on corporate sponsorship. R.P.B. has given expert testimony in relation to supplements and the brain. There was no role of funders in the conceptualization, design, data collection, analysis, decision to publish or preparation of the manuscript. The remaining authors declare no competing interests.
