## [Peer Review File · Nature Communications]

REVIEWER COMMENTS

Reviewer #1 (Remarks to the Author):

Mackenzie et al show that the majority of brain palmitate during development in mice is derived from hepatic DNL by using dietary sugars as carbon substrates. This conclusion was based on their special diet feeding enriched with different amounts of ^{13}C -depleted dietary palmitate and ^{13}C -enriched dietary sugars. While this approach is innovative and the conclusion is interesting, the interpretation of this study needs to be further supported by the key experiments suggested below.

1. For this reviewer, mUr is not an intuitive unit because a high negative value means low ^{13}C enrichment. Can the authors use $\% \text{ }^{13}\text{C}/(^{13}\text{C}+^{12}\text{C})$ enrichment throughout the manuscript for non-savvy readers? Or at least use both mUr and $\%$ enrichment?
2. For brain and liver lipids, the authors measured only phospholipids, but it is unclear how much these phospholipids contribute to the total lipid pools in these tissues. It is likely that other lipids such as neutral glycerides (MG, DG, TG) are abundant. Other lipids such as ceramides, cardiolipins, lysolipids and sphingolipids can be also a significant portion of the tissue lipids. The best way to check total fatty acid pools in these diverse lipid species would be saponification. Can the authors measure saponified fatty acid pool size and ^{13}C enrichment during development?
3. Among the lipogenic fatty acids, the authors only focused on palmitate, limiting the impact of this study. The authors explained how palmitate can be important for brain development but other lipogenic fatty acids can be equally important unless the authors show data that disproves this point. Enrichment and pool sizes of other lipogenic fatty acids such as palmitoleate, stearate and oleate should be measured. Also, since essential fatty acids like linoleate and arachidonic acid are only from a diet, their enrichment would be good to compare.
4. The authors claimed enhanced DNL based on RNA-seq data showing increased mRNA expression of lipogenic genes in liver but not in brain. However, lipogenic gene expression is too far away from lipogenic flux because flux is regulated by complex mechanisms including post-transcriptional, post-translational and allosteric regulations. Also, labeling from dietary sugar does not provide any quantitative information about DNL. The authors should measure increased DNL by using deuterated water tracing.

Reviewer #2 (Remarks to the Author):

This manuscript provides novel evidence on the mechanisms by which PAM levels are regulated in the brain during development. Although the overall manuscript is well-written and adds significant novelty to the current state of art in the field, I have the following concerns and suggested revisions.

Introduction:

- It is unclear what the main message is from the previously obtained results summed up between lines 96 to 103. Please add a conclusion or implication of what these results mean, what is the author's message here?
- The authors aim to investigate the origin of PAM during development after consumption of low, med, high PAM diets. Until the age of ~P14, the pups will not consume the PAM diets, but are fully fed by maternal milk. Therefore, it is crucial to know how the different levels of PAM in the diets are reflected in the milk of the dams? This information is needed to ensure that suckling pups indeed consume different levels of PAM.

Materials and Methods

- Currently, information on the amount of litters and pups per litter is missing from the paragraph "Animals" line 117, which creates a potential risk of between-litter variance. Please describe how many different litters are represented in each experimental group and how many pups per litter were included. If the number of litters included is different per outcome this needs to be described.
- Similarly, information on sex is completely missing in the Material & Methods section of the manuscript. Please include the male-to-female ratio per experimental group for all the outcomes. Are the experiments performed only in mice? Please provide a rationale for this sex-bias
- Please provide a rationale for the use of oleic acid as a lipid substitute for PAM in the LP diet to the paragraph "Diets" line 135.
- For the geotaxis and righting reflex test, please describe how many trials were performed per pup per day.

Results

- I cannot find the statistics for the statements in line 324 and 325, not in the text nor in the figure. Statistics must be provided to ensure "higher" and "slightly increased" are significant differences between the indicated groups.
- Statistics are missing in the paragraph between 326 and 332
- In lines 333-334, the authors state that tissue PAM levels were the same in females as in males. It is unclear which graph represents the males. If only males are presented in figure 1, it is important to state that this study is conducted only in male offspring, creating a sex-bias. If so, then the female relative PAM levels are around 50, while the male relative PAM levels are around 150, thus 3 times as high. If

these observations are correct, the authors cannot state that tissue PAM levels were the same in females as in males

- Line 361: “Differences in liver 13C-PAM between MP and HP were only..”

Discussion

- Please write the abbreviation of HFD in line 617 in full
- A general comment is that the neonatal neurodevelopment tasks are unlikely to pick up subtle differences in sensorimotor outcomes. I would advise more extensive behavioral tasks if the authors want to state that sensorimotor outcome is not affected. In addition, if the authors were interested in neurodevelopmental outcome, assessing the effects of the different diets on brain anatomy would have been a valuable addition to this manuscript.

Reviewer #3 (Remarks to the Author):

In this article Smith and colleagues present a research article that studies the origin of palmitic acid (PAM) in the brain (liver origin or brain origin) in different diets, containing low, medium and high levels of PAM and in different developmental stages. This was done by using a technique called compound specific isotope analysis, which allows the experimenters to discriminate between PAM present in the diet and PAM that was formed from glucose by the organism. They found that PAM levels in the brain were kept the same in the different time point regardless of dietary PAM. The liver saw an upregulation of de novo lipogenesis pathways suggesting pointing to compensatory mechanisms in the liver are responsible for the constant levels of PAM in the brain. They also performed RNA sequencing to identify the molecular pathways involved in the liver underlying such compensatory mechanisms.

General comments:

This manuscript is very well written. The use of the technique to answer a developmental question is extremely timely and relevant. The methods and results are very clear. The figures are very illustrative and the discussion is developed and complete.

If there was a small suggestion, I would add a few lines on developmental diseases where this technique could be useful to apply with the aim to expand our knowledge of diet and brain development.

Line 573: I would suggest changing the word virgin for non-bred mice, as it has a religious connotation and its use is discouraged in scientific writing.

Contents

Reviewer 1 (R1) Comments	2
R1: Comment 1	2
R1: Comment 2	2
R1: Comment 3	3
R1: Comment 4	6
R1: Comment 5	8
Reviewer 2 (R2) Comments:	11
R2: Comment 1	11
R2: Comment 2	11
R2: Comment 3	12
R2: Comment 4	13
R2: Comment 5	14
R2: Comment 6	15
R2: Comment 7	15
R2: Comment 8	15
R2: Comment 9	16
R2: Comment 10	16
R2: Comment 11	17
R2: Comment 12	17
R2: Comment 13	18
Reviewer 3 (R3) Comments:	19
R3 Comment 1:	19
R3 Comment 2:	19
R3 Comment 3:	19
R3 Comment 4:	20
References.....	21

Reviewer 1 (R1) Comments

R1: Comment 1

Mackenzie et al show that the majority of brain palmitate during development in mice is derived from hepatic DNL by using dietary sugars as carbon substrates. This conclusion was based on their special diet feeding enriched with different amounts of ¹³C-depleted dietary palmitate and ¹³C-enriched dietary sugars. While this approach is innovative and the conclusion is interesting, the interpretation of this study needs to be further supported by the key experiments suggested below.

Thank you for the comment. We have taken a considerable amount of time to revise, and therefore improve our manuscript in reply to your comments. Please see our point-by-point response below and addition of key experiments in reply to each of your concerns.

R1: Comment 2

For this reviewer, mUr is not an intuitive unit because a high negative value means low ¹³C enrichment. Can the authors use % ¹³C/(¹³C+¹²C) enrichment throughout the manuscript for non-savvy readers? Or at least use both mUr and % enrichment?

Thank you for the comment. We recognize this notation is not the most intuitive. The field adopted a universal δ notation system for reporting of natural abundance data because measurable differences in carbon-13 at the natural abundance level are very small (Brenna et al., 1997; Lacombe & Bazinet, 2020; McKINNEY et al., 1950). This notation reduces the redundant leading decimals that are attached to all natural abundance carbon isotope ratio (CIRs; ¹³C/¹²C) measurements (Lacombe & Bazinet, 2020). Because all natural abundance CIRs are expressed relative to a highly enriched primary reference standard (Vienna PeeDee Belemnite; VPDB) that defines the zero point for CIR measurements (0.01123720) (Coplen, 1995; Coplen et al., 2006a; Hoffman & Rasmussen, 2022), natural abundance CIRs possess a negative value when reported in the δ notation. Because the δ notation is dimensionless, the “Ur” was suggested for reporting CIRs to be compatible with the International System of Units (Brand & Coplen, 2012; Newell & Tiesinga, 2019).

A more thorough explanation of CIR notation was added to the Materials & Methods to assist the reader in interpretation of the data (page 11):

Isodat Workspace Version 3.0 (Thermo Scientific) was used to process chromatograms, in which tissue and dietary FAMES were identified by retention time relative to external reference material (Nu Chek Prep; item no. GLC-569). Using multipoint linear normalization, IRMS tissue and dietary carbon isotopic abundance data were normalized to the international carbon isotope ratio reference scale (Vienna PeeDee Belemnite; VPDB) (Coplen, 1995; Coplen et al., 2006b; Hoffman & Rasmussen, 2022). Each of USGS70, USGS71, and USGS72 certified reference material (Reston Stable Isotope Laboratories, United States Geological Services, Reston, VA) (Schimmelmann et al., 2016) were injected at least once per sequence and linear regression of the measured vs. true values were used to generate normalization equations. All R² values for tissue and dietary normalization equations were >0.999. Since measurable differences in ¹³C-values at the natural abundance level are small, ¹³C-values are reported according to the universal δ notation system in order to reduce multiple leading decimals attached to ¹³C values at the natural abundance level (Brenna et al., 1997; Lacombe & Bazinet, 2021; McKINNEY et al., 1950).

Additionally, since VPDM is highly enriched (0.01123720) (Hoffman & Rasmussen, 2022), all $\delta^{13}\text{C}$ values are negative; importantly, a less negative $\delta^{13}\text{C}$ value is more enriched in ^{13}C .

Additionally, % ^{13}C ($^{13}\text{C}/^{13}\text{C}+^{12}\text{C}$) was added as a background bar graph to main text figures containing CIRs to assist in interpreting enrichment (see updated Fig 1C, D and Figure 2C, D below):

Updated Figure 1C, D:

Updated Figure 2C, D:

R1: Comment 3

For brain and liver lipids, the authors measured only phospholipids, but it is unclear how much these phospholipids contribute to the total lipid pools in these tissues. It is likely that other lipids such as neutral glycerides (MG, DG, TG) are abundant. Other lipids such as ceramides, cardiolipins, lysolipids and sphingolipids can be also a significant portion of the tissue lipids. The best way to check total fatty acid pools in these diverse lipid species would be saponification. Can the authors measure saponified fatty acid pool size and ^{13}C enrichment during development?

Thank you for the comment. We agree that other neutral lipids are abundant in the liver. Therefore, we conducted thin layer chromatography to measure both levels of PAM, and $\delta^{13}\text{C}$ -PAM enrichment in liver triacylglyceride, phospholipid, cholesteryl ester, monoacylglycerol, diacylglycerol, and free fatty acid fractions at postnatal day 35 – as this was the timepoint for which RNAseq data was conducted and for which there was a sufficient amount of liver material left for analysis.

Levels (both concentration and relative percentage) of PAM in neutral lipid fractions of the liver at postnatal day 35 were added as new **Supplemental Figure 4A, B, C, D, E, and F** and referred to in the results section (page 17):

The difference in the level of total male pup liver PAM was most pronounced between pups fed the HP and LP diets and there was a significant dose-response reduction in liver PAM, in which lower dietary PAM resulted in significantly lower liver PAM (relative percentage: $p < 0.0001$; concentration: $p = 0.0024$). The significant dose-response reduction in the relative percentage, but not concentration, of total pup male liver PAM in response to diet persisted in separated liver neutral lipids at P35 including triacylglyceride (TAG) ($p < 0.0001$) and cholesteryl ester (CE) ($p = 0.0135$) fractions (Supplemental Figure 4A, C), however, not in separated PL, monoacylglycerol (MAG), diacylglycerol (DAG) and free fatty acid (FFA) fractions (Supplemental Figure 4B, D-F). Across all diets, the level of total male pup liver PAM peaked at P10 and significantly decreased at P21, and day 35 (relative percentage: $p < 0.0001$, concentration: $p = 0.0024$).

New Supplemental Figure 4A, B, C, D, E, F:

In addition to measuring levels of PAM in separated liver neutral lipids at P35, we also measured $\delta^{13}\text{C}$ -PAM enrichment in all mentioned liver neutral lipid fractions, which was added to the manuscript as new **Supplemental Figure 8A, B, C, D, E, and F** also referred to in the results section (page 19):

Differences in liver $\delta^{13}\text{C}$ -PAM between the LP and HP diet groups were the largest in which liver $\delta^{13}\text{C}$ -PAM was 16-31% significantly more enriched at all timepoints in male pups fed the LP compared to HP diet ($p < 0.0001$). Male pup liver $\delta^{13}\text{C}$ -PAM was also 10-24% significantly more enriched at all timepoints in male pups fed the LP diet, compared to the MP diet ($p < 0.0001$). The significant dose-response increase in total liver $\delta^{13}\text{C}$ -PAM in response to lower levels of dietary PAM in male pups persisted in the

separated male pup liver TAG fraction ($p = 0.0002$) (**Supplemental Figure 8A**), however, not in the separated male pup liver PL, CE, MAG, DAG, and FFA fractions at P35 (**Supplemental Figure 8B-F**).

Supplemental Figure 8A, B, C, D, E, F:

Given that the lipid composition of the brain is mostly glycerophospholipids (33% PC, 16% PE, PS 6%, SM 6%) (Yoon et al., 2022), and >98% of fatty acids in the brain are esterified into the glycerophospholipid fraction including PAM - one of the primary fatty acids representing brain PC, PE, and PS (Naudí et al., 2017) we measured total levels of PAM in the 5 major phospholipid fractions of the brain. This included measuring PAM levels in brain PC, PE, PI, PS and SM at all timepoints (**Supplemental Figure 11**) which compared to the liver were not influenced by diet at any timepoint.

This is reported in the results section (page 20):

Brain phospholipid class separation

Because total PAM levels were not different by diet in **male pup** brains, we explored if there may be a difference in **male pup** brain PAM levels in individual PL fractions. P0 and P10 were excluded from statistical analysis due to limited brain material. Nevertheless, levels of **male pup PAM in separated PL brain** fractions at P0 and P10 do not appear to be influenced by diet (**Supplemental Figure 11A, B**). As expected, at P21 and day 35 **male pup** PAM levels were significantly different by PL fraction ($p < 0.0001$ for all) (**Supplemental Figure 11C, D**), as levels of PAM in different PL pools are known to be different (Chen et al., 2008, 2013). Furthermore, there was no significant diet by PL interaction nor main effect of diet on PAM levels at P21 and day 35 in **male pups** (**Supplemental Figure 11C, D**), similar to the total **male pup** brain PAM finding. Interestingly, although CerPCho concentration appears low at all timepoints, the relative percentage of CerPCho decreases with time and appears highest at P0 in **male pups**.

Supplemental Figure 11:

R1: Comment 4

Among the lipogenic fatty acids, the authors only focused on palmitate, limiting the impact of this study. The authors explained how palmitate can be important for brain development but other lipogenic fatty acids can be equally important unless the authors show data that disproves this point. Enrichment and pool sizes of other lipogenic fatty acids such as palmitoleate, stearate and oleate should be measured. Also, since essential fatty acids like linoleate and arachidonic acid are only from a diet, their enrichment would be good to compare.

Thank you for the comment and we recognize the importance of examining other fatty acids, therefore, we have re-analyzed both the GC-FID the GC-C-IRMS chromatography and provided fatty acid levels, as well as fatty acid $\delta^{13}\text{C}$ -values and for all lipogenic (palmitoleate, stearate and oleate) and essential (linoleate and arachidonic acid) fatty acids, respectively, in the pups for which primary outcome data is presented. New data for the level of other fatty acids in pup brain and liver can be found in new **Supplemental Figure 6A, B, C, D, E, F** and **7A, B, C, D, E, F**, respectively. New data for the $\delta^{13}\text{C}$ enrichment of other fatty acids in pup brain and liver can be found in new **Supplemental Figure 9A, B, C, D, E, F** and **10A, B, C, D, E, F**, respectively.

New **Supplemental Figures 6A, B, C, D, E, F** and **7A, B, C, D, E, F** were referred to in the results section (page 17-18):

*Similar to male pup tissue levels of PAM, there was not a significant main effect of diet, only a significant main effect of time ($p < 0.0001$ for all) in the concentration of other male pup lipogenic fatty acid acids in the brain including 16:1, 18:0 and OLA (**Supplemental Figure 6A, B, C**). In contrast, there was a significant diet x time interaction in the concentration ($p = 0.0036$) and relative percentage ($p = 0.0001$) of liver 16:1 (**Supplemental Figure 7A**), in which male pups fed the LP diet had significantly lower concentration of liver 16:1 than male pups fed the MP and HP diet at P0, but not P10-35. However, male pups fed the LP diet also had a significantly lower relative percentage of liver 16:1 ($p < 0.0001$) than mice fed the HP diet at all timepoints examined (**Supplemental Figure 7A**). Additionally, there was a significant main effect of diet in liver OLA whereby male pups fed the LP diet (containing high OLA) had a higher concentration ($p = 0.0005$) and relative percentage ($p < 0.0001$) of liver OLA than male pups fed the MP and HP diet (containing equal parts of dietary OLA to PAM, and low dietary OLA, respectively)*

(Supplemental Figure 7C). Lastly, there was not a significant effect of diet, only a significant main effect of time ($p < 0.0001$ for all) in the concentration of other n-6 fatty acids including 18:2n-6 and 20:4n-6 in the brain (Supplemental Figure 6D, E) and the liver (Supplemental Figure 7D, E). For visual purposes, only liver 16:1 concentration means significantly different by diet at each timepoint are displayed in Supplemental Figure 7A.

New Supplemental Figure 6A, B, C, D, E, F:

New Supplemental Figure 7A, B, C, D, E, F:

New Supplemental Figures 9A, B, C, D, E, F and 10A, B, C, D, E, F were referred to in the results section (page 19-20):

Similar to male pup brain $\delta^{13}\text{C}$ -PAM, there was a significant diet x time interaction in other male pup lipogenic brain and liver fatty acid $\delta^{13}\text{C}$ values, including brain $\delta^{13}\text{C}$ -18:0 ($p = 0.0036$) (Supplemental Figure 9B) and liver $\delta^{13}\text{C}$ -18:0 ($p < 0.0001$) (Supplemental Figure 10B), as well as brain $\delta^{13}\text{C}$ -OLA ($p = 0.0398$) (Supplemental Figure 9C). Additionally, there was a significant main effect of diet in brain ($p < 0.0001$) and liver ($p < 0.0001$) $\delta^{13}\text{C}$ -16:1 (Supplemental Figure 9A and 10A, respectively), whereby male pups fed the LP diet had more enriched $\delta^{13}\text{C}$ -16:1, compared to male pups fed the MP and HP diets in both tissues. Although male pup liver $\delta^{13}\text{C}$ -18:0 depended on time, similar to male pup liver $\delta^{13}\text{C}$ -PAM, liver $\delta^{13}\text{C}$ -16:1 and $\delta^{13}\text{C}$ -18:0 had also had a dose-response increase in $\delta^{13}\text{C}$ in response to low dietary PAM (Supplemental Figure 10A, B). In contrast, the significant main effect of diet in male pup liver $\delta^{13}\text{C}$ -OLA ($p < 0.0001$) showed the reverse; a dose-response increase in $\delta^{13}\text{C}$ -OLA in response to high dietary PAM (and low dietary OLA) (Supplemental Figure 10C). Like male pup tissue $\delta^{13}\text{C}$ -PAM, any differences between diet groups were more pronounced overall at the level of the liver, compared to the brain for $\delta^{13}\text{C}$ -16:1, -18:0 and -OLA (Supplemental Figure 10A, B, C). Interestingly, compared to male pup brain

$\delta^{13}\text{C}$ -PAM and -16:1 which displayed a similar dose response increase in $\delta^{13}\text{C}$ in response to low dietary PAM as liver $\delta^{13}\text{C}$ -PAM and -16:1, differences in male pup brain $\delta^{13}\text{C}$ -18:0 and -OLA were less clear (Supplemental Figure 9B, C) and did not follow a similar response as liver $\delta^{13}\text{C}$ -18:0 and -OLA (Supplemental Figure 10B, C). Importantly, male pup $\delta^{13}\text{C}$ values of other n-6 fatty acids 18:2n-6 and 20:4n-6, as well as n-3 fatty acid DHA, were not affected by diet in either the brain (Supplemental Figure 9D-F) or the liver (Supplemental Figure 10D-F).

New Supplemental Figure 9A, B, C, D, E, F:

New Supplemental Figure 10A, B, C, D, E, F:

R1: Comment 5

The authors claimed enhanced DNL based on RNA-seq data showing increased mRNA expression of lipogenic genes in liver but not in brain. However, lipogenic gene expression is too far away from lipogenic flux because flux is regulated by complex mechanisms including post-transcriptional, post-translational and allosteric regulations. Also, labeling from dietary sugar does not provide any quantitative information about DNL. The authors should measure increased DNL by using deuterated water tracing.

Thank you for the comment. While we agree that measuring increased DNL using deuterated water tracing would provide further validation of measuring increased DNL using carbon-13, we wanted to bring to your attention a recent paper published in Nature Communications in June of 2021 (Fu et al., 2021): <https://www.nature.com/articles/s41467-021-23958-4>.

This paper elegantly measured lipogenic flux specifically in response to nutritional interventions by high-resolution Orbitrap gas-chromatography mass-spectrometry (HR-Orbitrap-GC-MS), resolving both ^2H and ^{13}C simultaneously. The results of Fu et al. showed that their HR-Orbitrap-GC-MS method was

capable of detecting significant differences in lipogenic flux after fasting and refeeding protocols in both mice (Figure 4a) and humans (Figure 4d) following an administered dose of $^2\text{H}_2\text{O}$. Furthermore, authors compared their method of HR-Orbitrap-GC-MS detection of ^2H -PAM with ^3H radioactivity in liver TAGs of SREBP-1a transgenic mice after a dose of either $^2\text{H}_2\text{O}$ or $^3\text{H}_2\text{O}$, respectively. Results obtained using HR-Orbitrap-GC-MS closely matched the parallel experiment performed using $^3\text{H}_2\text{O}$ (Fig 4b).

Importantly, although techniques are increasingly being developed to measure isotopes at the natural abundance level with greater precision including HR-Orbitrap-GC-MS (Eiler et al., 2017), gas chromatography-isotope ratio mass spectrometry (GC-IRMS) remains the most widely used analytical technique for such measurements due to its high precision. For reference, changes at the natural abundance level in carbon isotope ratios that are of relevance to biologists and in our case, nutritional interventions, are +1 mUr (equivalent to +0.001 atom % excess) (Lacombe & Bazinet, 2021). This precision necessitates the use of GC-IRMS for our study, which is sensitive below 0.001 atom % excess. Furthermore, depending on the abundance of the analyte GC-IRMS can measure natural abundance isotope ratios with accuracies of 4-6 significant digits (Lacombe & Bazinet, 2021).

Fu et al. cites that both hydrogen isotopes (^2H and ^3H) as well as carbon isotopes (^{13}C and ^{14}C) are candidate tracers of lipogenesis (Brunengraber et al., 1997; Murphy, 2006; Wallace & Metallo, 2020); specifically under the condition that lipogenic substrates that form acetyl-CoA (i.e. glucose) are capable of detecting lipogenesis with high sensitivity when the acetyl-CoA pool is not diluted by other acetyl-CoA donors (Fu et al., 2021). Accordingly, our study utilized a controlled animal diet with known $\delta^{13}\text{C}$ -values of dietary macronutrients capable of supplying acetyl-CoA including dietary sugars (-11.15 ± 0.65 mUr), dietary fat (-28.26 to -29.70 mUr), and protein (-23.88 ± 0.12 mUr). Therefore, the combination of known dietary inputs and highly precise methodology utilized in our study allow for the accurate measurement of lipogenesis.

Overall, given the impracticality of feeding $^2\text{H}_2\text{O}$ to weanling mice alongside the chronic nature (long term feeding intervention) of our study, if we were to conduct a study to validate our carbon-13 result using $^2\text{H}_2\text{O}$ we would opt for a fasting and refeeding experiment in postnatal mice. If this were the case, we feel our study design and our result would be strikingly similar to the study by Fu et al. published in Nature Communications, and arguably, executed with even more precision by GC-IRMS as compared to HR-Orbitrap-GC-MS.

Accordingly, we kindly ask the reviewer to consider the combination of **1)** known dietary isotopic inputs as well as **2)** high precise methodology utilized in our study, alongside **3)** the study by Fu et al. published in Nature Communications before proceeding with the reviewer request to validate our technique using $^2\text{H}_2\text{O}$.

Nevertheless, because of the complexities surrounding our methodology, we have added new **Supplemental Figure 2** to the manuscript to better explain the sources capable of maintaining tissue PAM pools, which give rise to tissue $\delta^{13}\text{C}$ -PAM values measured in our study (page 8):

*The weighted average of all carbohydrate sources in the three diets was measured to be more enriched in ^{13}C (-11.15 ± 0.65 mUr), than dietary PAM (-29.44 ± 0.12 mUr to -29.70 ± 0.19 mUr), OLA (-28.26 ± 0.12 mUr to -28.39 ± 0.12 mUr) or casein (-23.88 ± 0.12 mUr) making dietary carbohydrates distinguishable from dietary fat and protein isotopically (**Table 1**). **The importance of dietary***

macronutrients being distinguishable isotopically is illustrated in **Supplemental Figure 2**, as it allows for the measurement of tissue $\delta^{13}\text{C}$ -PAM which gives rise to fatty acid origin.

New Supplemental Figure 2:

Lastly, reviewer 1 advised us to either tone down our argument or discuss limitations appropriately in our manuscript, as our method used does not directly support total lipogenesis flux changes.

Accordingly, we have added 2 limitation and future direction sections in our manuscript surrounding **A)** the RNAseq data (page 27):

Nevertheless, a limitation of our work includes the fact lipogenic genes are regulated by complex mechanisms, for example, by transcription factors sterol regulatory element binding protein-1c and carbohydrate element-binding protein - which are responsive to cell signaling and intermediates of glycolysis (Fu et al., 2021; Horton et al., 2002; Ortega-Prieto & Postic, 2019). Therefore, in addition to gene expression, future studies should measure post-transcriptional, post-translational, and allosteric regulators of lipogenesis to gain a more holistic understanding of lipogenesis in response to diet.

And **B)** the GC-C-IRMS data (page 28):

Future studies should characterize the contribution of other molecules that contribute to tissue acetyl-CoA pools used in synthesizing PAM during development in addition to dietary sugars, as a limitation of our work is the inability to characterize and quantify other molecules which contribute to the depletion of brain $\delta^{13}\text{C}$ -PAM, other than dietary fatty acids and protein measured here. Furthermore, future studies could measure hydrogen isotopes of water incorporated into PAM sourced from lipogenesis, which enables quantification of the fraction of newly synthesized PAM.

Lastly, we have softened language throughout the manuscript when referring to the upregulation of DNL, specifying it is only the upregulation of genes involved in DNL. For example, in our conclusion (page 32):

Overall, our study is the first to show the upregulation of genes involved in hepatic DNL from dietary sugar to supply the male murine brain with PAM, which was augmented in response to low levels of dietary PAM fed from birth. We also show brain CIRs of PAM are responsive to levels of dietary PAM during the pre- and post-weaning period, putting forward a novel and feasible measure of tissue PAM origin and homeostasis during development, under the condition dietary precursors and products are well-defined. Importantly, in response to low dietary PAM, we show that there is compensation in both the dam and the pup to ensure total brain PAM pools are maintained by the upregulation of fatty acid

synthesis by peripheral tissues, suggesting the importance of PAM regulation during the postnatal period.

In our abstract (page 3):

Conclusions: *DNL from dietary sugars maintains the majority of brain PAM during development and is augmented in mice fed low PAM from birth. Importantly, the upregulation of genes involved in hepatic DNL from dietary sugars in response to low PAM identifies a compensatory mechanism to maintain total brain PAM pools compared to periphery - which ultimately suggests an importance of brain PAM regulation during development.*

In our introduction (page 6):

We found the majority of brain PAM was maintained by DNL from dietary sugars, augmented in mice fed low PAM from birth. Furthermore, we identified genes involved in hepatic DNL from dietary sugars were upregulated in mice fed low compared to high PAM at day 35 - a compensatory mechanism identified to supply the brain with PAM during development.

Reviewer 2 (R2) Comments:

R2: Comment 1

This manuscript provides novel evidence on the mechanisms by which PAM levels are regulated in the brain during development. Although the overall manuscript is well-written and adds significant novelty to the current state of art in the field, I have the following concerns and suggested revisions.

Thank you for the comment and for recognition of the novelty of our work.

R2: Comment 2

It is unclear what the main message is from the previously obtained results summed up between lines 96 to 103. Please add a conclusion or implication of what these results mean, what is the author's message here?

Thank you for the comment, we recognize this previously may not have been clear. The importance of stating these results lies in the fact our technique (CSIA) to study brain PAM origin has only been utilized in adult mouse models and has not yet been utilized during development to examine brain PAM origin – a time when accretion of PAM to the developing brain is rapid. A statement was added to the end of these results in the introduction to better connect the previous work to the current work (page 5):

Our group recently utilized CSIA to study brain PAM origin and found approximately 70% of the brain PAM pool is maintained by DNL from dietary sugars, 44% of which was derived from local DNL within the brain in response to feeding standard levels of dietary PAM to adult mice (8%) (Lacombe et al., 2018). Furthermore, our group utilized CSIA and found 69–80% and 47–58% of the brain PAM pool was maintained by total and local brain DNL from dietary sugars, respectively, and DNL from dietary sugars was augmented in adult mice fed low PAM (<2%) levels from birth compared to a high PAM intervention (>95%) (M. E. Smith et al., 2022). However, brain PAM origin during development has not been investigated utilizing CSIA; a time when PAM is rapidly accreted to the brain (Clandinin et al., 1980a, 1980b).

R2: Comment 3

The authors aim to investigate the origin of PAM during development after consumption of low, med, high PAM diets. Until the age of ~P14, the pups will not consume the PAM diets, but are fully fed by maternal milk. Therefore, it is crucial to know how the different levels of PAM in the diets are reflected in the milk of the dams? This information is needed to ensure that suckling pups indeed consume different levels of PAM.

Thank you for the comment. We agree that knowledge of pre-weaning levels of PAM consumption is crucial for this study design. The stomach content of a wild-type weanling pup or “milk spot” (image from our study below) reflects pre-weaning mouse diet (Kim et al., 2014; Turgeon & Meloche, 2009).

Accordingly, we collected P0 and P10 stomach content, extracted lipids, and ran the samples by gas chromatography-flame ionization detection to determine PAM levels. This data can be found in **Supplemental Figure 3** – in which the levels of PAM (both $\mu\text{mol/g}$ stomach content and relative percentage) in weanling pup stomach content reflects dietary PAM levels; whereby weanling pups fed low PAM were found to have significantly lower stomach PAM than weanling pups fed medium, and high PAM diets at P0 ($p < 0.0001$). Because the examination of the milk spot depended on the fed state of euthanized pups, a smaller sample size at P10 (LP = 2, MP = 5, HP = 5) did not allow for statistical analysis. Nevertheless, both concentration and relative percentage of PAM reflect dietary levels of PAM and follow the same trend of P0.

We recognize this may not have been clear, therefore, we have added a statement to the methods section outlining how pre-weaning dietary PAM levels were assessed (page 8):

*Because weanling pups consumed dam milk from P0-P21, the fatty acid profile of P0 and P10 stomach content was assessed to ensure pups also consumed different levels of PAM pre-weaning. Importantly, P0 and P10 stomach content also contained low (21-29%), medium (25-35%), and high (31-41%) PAM levels (**Supplemental Figure 3**) reflecting levels of dietary PAM. Feeding of the diets was performed by one investigator (MES); therefore, blinding to the study diets was not possible.*

Supplemental Figure 3:

R2: Comment 4

Currently, information on the amount of litters and pups per litter is missing from the paragraph “Animals” line 117, which creates a potential risk of between-litter variance. Please describe how many different litters are represented in each experimental group and how many pups per litter were included. If the number of litters included is different per outcome this needs to be described.

Thank you for the comment. Information on the number of total litters, pups per litter, as well as pup sex per litter were added as a new table; **Supplemental Table 1**. We reference this new data under “Animals”, in addition to stating the total number of litters born (n = 29), as well as the exclusion of a litter (n = 2) from the study due to infanticide (page 6):

*Of the 32 bred dams, 29 dams (91%) had litters at the end of gestation, however, 2 dams and their litters were excluded due to the occurrence of dam infanticide (**Supplemental Table 1**) to reduce between-litter variance.*

As between-litter variance is a concern, we also analyzed the litter data and confirmed the total number of pups born per dam (p = 0.5131), male pups born per dam (p = 0.6164), or female pups born per dam (p = 0.4621) were not statistically different (also added as new **Supplemental Figure 1A, B, C, D**). It should also be noted we did confirm that there were no significant differences in maternal behaviour, assessed through nest building, between dams fed different diets (**Supplemental Figure 11D**). Lastly, the ratio of male: female pups was not different by diet (p = 0.4574). Therefore, litter effects known to impact results in developmental studies, specifically behavioural studies, are not likely (Williams et al., 2017). This new data was also added to the “Animals” section (page 6-7):

*Importantly, there were no significant differences between diet groups in the total number of pups, nor the total number of female or male pups born per dam (**Supplemental Figure 1**). Nevertheless, although the ratio of male:female pups born was not different by diet, all dams produced a significantly higher number of male pups, compared to female pups (p < 0.0001) (**Supplemental Figure 1**) in our study.*

Because we cannot pool siblings from the same litter, pups were selected at each timepoint from different litters. Therefore, for each experimental group, the number of pups directly represents the

number of litters as described in new **Supplemental Table 1**. All litters were utilized throughout the study, and number of litters represented per outcome differed due to dependence on total number of pups born, as well as material available per assay. This was added to “Animals” section as well, and is also outlined in **Supplemental Table 1** (page 7):

Same-sex pup siblings were not pooled for outcomes and the distribution of litter representation across outcomes is reported in Supplemental Table 1.

New Supplemental Figure 1A, B, C, D:

R2: Comment 5

Similarly, information on sex is completely missing in the Material & Methods section of the manuscript. Please include the male-to-female ratio per experimental group for all the outcomes. Are the experiments performed only in mice? Please provide a rationale for this sex-bias

Thank you for the comment. In our study, the total number of male pups born per litter was significantly higher than female pups born per litter ($p < 0.0001$) (new **Supplemental Figure 1A, B, C, D**). Litters in our study were born throughout the summer (July and August), and seasonal variation is known to impact litter sex ratio; in which both captive wild and laboratory mice produce more male- than female- biased litters in the spring and summer months (Drickamer, 1990). Importantly, diets high or low in saturated fats have been shown to impact male:female litter ratios (Rosenfeld et al., 2003) and we confirm male:female litter ratios in our study were not due to any differences in diet group ($p = 0.4574$).

Therefore, the rationale for sex-bias in our study was due to limited availability of females at each timepoint. Nevertheless, we collected a subset of females ($n = 1-3$) females per timepoint, per diet, when possible to supplement the male data. The only case in which male and female data are pooled data in our study is for P0-10 behavioural tests to increase sample size for the nature of statistically assessing behaviour as stated in the figure legend (**Supplemental Figure 12A and B**). Otherwise, the number of males to females utilized for all other outcomes can now be found in new **Supplemental Table 1**. The rationale for the sex-bias in our study, as well as where to locate male: female ratios for experimental outcomes has been added under the “Animals” section (page 6-7):

Nevertheless, although the ratio of male:female pups born was not different by diet, all dams produced a significantly higher number of male pups, compared to female pups ($p < 0.0001$) (Supplemental Figure 1)

in our study. Therefore, our primary study outcomes (¹³C data and RNAseq) were powered to assess male pups, and female pups were included, when possible, to supplement male data (Supplemental Table 1).

R2: Comment 6

Please provide a rational for the use of oleic acid as a lipid substitute for PAM in the LP diet to the paragraph “Diets” line 135.

Thank you for the comment. Our goal in using oleic acid as a lipid substitute for PAM in the LP diet is because oleic acid is not a precursor to PAM, and rather, a downstream product of PAM after elongation and desaturation. Therefore, using oleic acid as a lipid substitute for PAM in the LP diet does not allow for the downstream production of PAM. Additionally, our group has previously utilized oleic acid as a replacement in diets for which the total fatty acid composition must be balanced (Giuliano et al., 2018; Hopperton et al., 2016; M. E. Smith et al., 2022).

This information was added to the “Diets” section of the methods for clarity (page 7):

The three study diets were isocaloric (% by calorie: 16.8% fat, 19.4% protein and 63.9% carbohydrates), differing only in fatty acid composition and modified from the AIN-93G rodent diet. Diets were formulated at Dyets Inc. using palmitate and oleate ethyl ester (Nu Chek Prep Inc., item no. N-16-E, U-46-E). The LP (item no. 104636) and HP (item no. 104638) diets contained 15.6% energy of either oleic acid (18:1n-9; OLA), or PAM, respectively, while the MP diet (item no. 104637) contained 7.8% energy from PAM, and 7.8% energy from OLA. As OLA has been previously utilized by our group as a fatty acid substitute to achieve matched total fat across different animal diets (Giuliano et al., 2018; Hopperton et al., 2016; M. E. Smith et al., 2022) and is a downstream product of PAM rather than a precursor, it was used as a substitute for PAM in the LP diet

R2: Comment 7

For the geotaxis and righting reflex test, please describe how many trial were performed per pup per day.

Thank you for the comment. In order to reduce time away from the dam while utilizing the same investigator per timepoint (P0, P2, P4, P6, P8, P10) trials were performed with an inter-test time of 1 hour. This has been added to the material and methods section (page 16):

Pup reflexes were evaluated between the hours of 09:00 and 12:00 by one investigator (MES) for 6 consecutive days (P0, P2, P4, P6, P8, P10) using the geotaxis and righting reflex tests with an inter-test time of 1 hour to ensure diets did not impact pup sensorimotor development using methods adapted from other groups (Arsenault et al., 2014; Heyser, 2003).

R2: Comment 8

I cannot find the statistics for the statements in line 324 and 325, not in the next nor in the figure. Statistics must be provided to ensure “higher” and “slightly increased” are significant differences between the indicated groups.

Thank you for the comment. Statistics (p-values) were initially not included in lines 324 and 325 because there was not a significant diet x time interaction ($p > 0.05$) in the relative percentage of pup brain PAM, therefore, multiple comparisons are not possible. Nevertheless, although we stated the main effect of time ($p < 0.0001$) in lines 321-322, we have re-stated this p-value in the to the statements in lines 324

and 325 to provide a higher level of detail regarding the differences between groups. Additionally, when the word “higher” is used in the results section, it has been replaced with “significantly higher” and supplemented with a p-value to ensure the reader there are indeed significant differences between groups. Furthermore, words like “slightly increased” have been replaced with “significantly increased” and also supplemented with a p-value in this statement, and throughout the manuscript. For example (page 16):

Across all diets, the relative percentage of male pup brain PAM peaked at P10, significantly higher than at P21 and day 35 ($p < 0.0001$). As opposite, the concentration of male pup brain PAM significantly increased with time from P10 to P21 and day 35 ($p < 0.0001$).

R2: Comment 9

Statistics are missing in the paragraph between 326 and 332

Thank you for the comment. Similar to lines 324-325, statistics (p-values) were initially not included in lines 326-332 because there was not a significant diet x time interaction ($p > 0.05$) in the relative percentage of pup liver PAM, therefore, multiple comparisons are not possible. Nevertheless, although we stated the main effect of time ($p < 0.0001$) and diet ($p < 0.0001$) in lines 322-323, we have re-stated this p-value in the to the statements in lines 326-332 to provide a higher level of detail regarding the differences between groups. For example (page 17):

The difference in the level of total male pup liver PAM was most pronounced between pups fed the HP and LP diets and there was a significant dose-response reduction in liver PAM, in which lower dietary PAM resulted in significantly lower liver PAM (relative percentage: $p < 0.0001$; concentration: $p = 0.0024$).

R2: Comment 10

In lines 333-334, the authors state that tissue PAM levels were the same in females as in males. It is unclear which graph represents the males. If only males are presented in figure 1, it is important to state that this study is conducted only in male offspring, creating a sex-bias. If so, then the female relative PAM levels are around 50, while the male relative PAM levels are around 150, thus 3 times as high. If these observations are correct, the authors cannot state that tissue PAM levels were the same in females as in males

Thank you for the comment. In our study, dams had a significantly higher number of male compared to female pups within their litters (see new **Supplemental Figure 1A, B, C, D** new **Supplemental Table 1**). Therefore, females were only taken when possible to supplement primary outcome male data – for which the primary outcomes were powered with. For clarity, this has been added to the methods section as stated previously (page 6-7):

*Nevertheless, although the ratio of male:female pups born was not different by diet, all dams produced a significantly higher number of male pups, compared to female pups ($p < 0.0001$) (**Supplemental Figure 1**) in our study. Therefore, our primary study outcomes (^{13}C data and RNAseq) were powered to assess male pups, and female pups were included, when possible, to supplement male data.*

As for the comparison between male and female pups in the relative percentage of pup brain PAM, we apologize, as we recognize our previous right y-axis on Figure 1 (males) went up to 100%, while our right y-axis on Supplemental Figure 3 (females) went up to 40% making comparisons difficult. Furthermore,

the numbers “50” and “150” you are referring to represent concentration ($\mu\text{mol/g}$) (left y-axis) and not the relative percentage of PAM (right y-axis). After ensuring both right y-axes are to scale, you can see there are no such large differences between female pups (left figure) and male pups (right figure) below. These figures (**Supplemental Figure 5A** and **Figure 1A**) have been updated accordingly.

Nevertheless, we have lightened our language since our sample size for female mice is not large enough to conduct statistics between makes and females. Words like “same” throughout the manuscript when making such comparisons have been replaced with “similar to”. For example (page 19):

*We observed a subset of female pup tissue $\delta^{13}\text{C}$ -PAM was similar to that of male pups fed the LP, MP, and HP diets from birth (**Supplemental Figure 5C, D**).*

Lastly, we have updated figure titles, legends, results, and discussion to mention “males” or “females” when reporting an outcome. For example (page 25):

Despite feeding diets vastly different in PAM composition to pups, total male pup brain PAM was maintained at all timepoints directly contrasting the level of PAM in the male pup liver where levels reflected dietary intake of PAM. Furthermore, the concentration of PAM in a subset of male pup brain samples was maintained in all individual PL fractions. Therefore, male pup brain $\delta^{13}\text{C}$ -PAM was examined to determine the sources of PAM maintaining this pool.

R2: Comment 11

Line 361: “Differences in liver ^{13}C -PAM between MP and HP were only..”

Thank you for the comment. Similar to comments 9 and 10 we have added a p-value for the main effect of diet ($p < 0.0001$) to all conclusions and avoided making between-group comparisons at each timepoint (page 19):

*To investigate the dietary origin of sources maintaining the pup liver pool, we measured male pup liver $\delta^{13}\text{C}$ -PAM. There was a significant main effect of diet ($p < 0.0001$) and time ($p < 0.0001$) in male pup liver $\delta^{13}\text{C}$ -PAM (**Figure 1D**). Across all timepoints, there was a significant dose-response increase in male pup liver $\delta^{13}\text{C}$ -PAM in response to lower levels of dietary PAM similar to the brain, that was more pronounced. Differences in liver $\delta^{13}\text{C}$ -PAM between the LP and HP diet groups were the largest in which liver $\delta^{13}\text{C}$ -PAM was 16-31% significantly more enriched at all timepoints in male pups fed the LP compared to HP diet ($p < 0.0001$).*

R2: Comment 12

Please write the abbreviation of HFD in line 617 in full

Thank you for the comment. The abbreviation of HFD was written out in full at first use (page 31):

It has been demonstrated that maternal exposure to a high-fat diet (HFD) compared to a control chow diet results in delayed physical and sensorimotor maturation, namely, delays in righting reflexes (Giriko et al., 2013; Souto et al., 2020; Teo et al., 2017).

R2: Comment 13

A general comment is that the neonatal neurodevelopment tasks are unlikely to pick up subtle differences in sensorimotor outcomes. I would advise more extensive behavioral tasks if the authors want to state that sensorimotor outcome is not affected. In addition, if the authors were interested in neurodevelopmental outcome, assessing the effects of the different diets on brain anatomy would have been a valuable addition to this manuscript.

Thank you for the comment. While we agree more extensive behavioural tasks would be useful in concluding that sensorimotor outcomes are not affected by diet, such extensive behavioural testing in developing rodents is limited; the eyes and ear canals remain closed for several days after birth and tactile sensitivity is not fully developed (Heyser, 2003). Therefore, the most commonly used protocols for investigating behaviour in rodents during development are tests of developmental reflexology including negative geotaxis and righting reflex (Heyser, 2003), which were used in our study.

Additionally, we tested our pups at postnatal days 0, 2, 4, 6, 8 and 10. Although testing time was limited to 120 seconds a day per pup, removing the pup from the dam causes a significant amount of maternal stress and negatively impact maternal behaviours (Orso et al., 2018). Therefore, we opted to minimize the separation time between the pups and dams to avoid maternal stress; since cases of infanticide were already observed in our study we believe more tests would have exacerbated this issue.

Nevertheless, the reason we conducted the behaviour was because our diets had never been used before and we wanted to ensure our diets were not interfering with what would be considered normal development. Therefore, we did not make any conclusions throughout the manuscript about neurodevelopmental outcomes. Nevertheless, we have gone through the manuscript and been more careful with our language/conclusion surrounding the behavioural data.

For example, page 5-6:

To the best of our knowledge our diets have not been utilized during development, therefore, we also conducted maternal behaviour tests during gestation, as well as sensorimotor development tests in the pups during the first 10 days of life to ensure our diets were not impacting behaviour. We found the majority of brain PAM was maintained by DNL from dietary sugars, augmented in mice fed low PAM from birth. Furthermore, we identified genes involved in hepatic DNL from dietary sugars were upregulated in mice fed low compared to high PAM at day 35 - a compensatory mechanism identified to supply the brain with PAM during development. Dietary PAM levels did not have a significant effect on maternal behaviour or pup sensorimotor development.

For example, page 16:

To ensure diets did not impact pup sensorimotor development, pup reflexes were evaluated between the hours of 09:00 and 12:00 by one investigator (MES) for 6 consecutive days (P0, P2, P4, P6, P8, P10) using the geotaxis and righting reflex tests with an inter-test time of 1 hour using methods adapted from other groups (Arsenault et al., 2014; Heyser, 2003).

Page 32:

Therefore, we investigated if isocaloric diets with a standard percentage of energy from fat, composed of different PAM compositions would also affect physical and sensorimotor maturation because we wanted to **ensure our diets were not interfering with behaviour**, which to the best of our knowledge has not been explored.

Reviewer 3 (R3) Comments:

R3 Comment 1:

In this article Smith and colleagues present a research article that studies the origin of palmitic acid (PAM) in the brain (liver origin or brain origin) in different diets, containing low, medium and high levels of PAM and in different developmental stages. This was done by using a technique called compound specific isotope analysis, which allows the experimenters to discriminate between PAM present in the diet and PAM that was formed from glucose by the organism. They found that PAM levels in the brain were kept the same in the different time point regardless of dietary PAM. The liver saw an upregulation of de novo lipogenesis pathways suggesting pointing to compensatory mechanisms in the liver are responsible for the constant levels of PAM in the brain. They also performed RNA sequencing to identify the molecular pathways involved in the liver underlying such compensatory mechanisms.

Thank you for the comment and thorough summary of our work.

R3 Comment 2:

This manuscript is very well written. The use of the technique to answer a developmental question is extremely timely and relevant. The methods and results are very clear. The figures are very illustrative and the discussion is developed and complete.

Thank you for the comment and for the recognition of the relevance as well as timeliness of our work, we appreciate it.

R3 Comment 3:

If there was a small suggestion, I would add a few lines on developmental diseases where this technique could be useful to apply with the aim to expand our knowledge of diet and brain development.

Thank you for the great idea. We added a statement to the discussion for which our technique could be relevant in studying both developmental, and degenerative diseases of the brain (page 31):

*Although we did not have enough material to measure liver $\delta^{13}\text{C}$ -cholesterol, this measure would be feasible for future studies to test this hypothesis and explore cholesterol origin in response to low and high PAM feeding. **Furthermore, on a broader scale, our technique could be applied in studying developmental or degenerative disorders to better understand the relationship between diet and the developing or aging brain. For instance, measuring $\delta^{13}\text{C}$ values of cholesterol precursors in the study of inborn errors of cholesterol biosynthesis, or, measuring $\delta^{13}\text{C}$ values of brain fatty acids to study lower***

glucose uptake, glucose hypometabolism and glucose hypermetabolism in the case of diabetes, cancer, and Alzheimer's disease, respectively.

R3 Comment 4:

Line 573: I would suggest changing the word virgin for non-bred mice, as it has a religious connotation and its use is discouraged in scientific writing.

Thank you for the comment, this was an oversight on our end. The word “virgin” has been changed to “non-bred” mice (page 29):

Importantly, the study by S. Smith et al., 1969 also found the lipogenic capacity of the liver increased during lactation compared to non-bred mice, and the liver, but not the mammary gland had altered activity of enzymes involved in DNL (S. Smith et al., 1969).

References

- Arsenault, D., St-Amour, I., Cisbani, G., Rousseau, L.-S., & Cicchetti, F. (2014). The different effects of LPS and poly I:C prenatal immune challenges on the behavior, development and inflammatory responses in pregnant mice and their offspring. *Brain, Behavior, and Immunity*, *38*, 77–90. <https://doi.org/10.1016/j.bbi.2013.12.016>
- Brand, W. A., & Coplen, T. B. (2012). Stable isotope deltas: Tiny, yet robust signatures in nature. *Isotopes in Environmental and Health Studies*, *48*(3), 393–409. <https://doi.org/10.1080/10256016.2012.666977>
- Brenna, J. T., Corso, T. N., Tobias, H. J., & Caimi, R. J. (1997). High-precision continuous-flow isotope ratio mass spectrometry. *Mass Spectrometry Reviews*, *16*(5), 227–258. [https://doi.org/10.1002/\(SICI\)1098-2787\(1997\)16:5<227::AID-MAS1>3.0.CO;2-J](https://doi.org/10.1002/(SICI)1098-2787(1997)16:5<227::AID-MAS1>3.0.CO;2-J)
- Brunengraber, H., Kelleher, J. K., & Des Rosiers, C. (1997). Applications of mass isotopomer analysis to nutrition research. *Annual Review of Nutrition*, *17*, 559–596. <https://doi.org/10.1146/annurev.nutr.17.1.559>
- Chen, C. T., Domenichiello, A. F., Trépanier, M.-O., Liu, Z., Masoodi, M., & Bazinet, R. P. (2013). The low levels of eicosapentaenoic acid in rat brain phospholipids are maintained via multiple redundant mechanisms. *Journal of Lipid Research*, *54*(9), 2410–2422. <https://doi.org/10.1194/jlr.M038505>
- Chen, C. T., Ma, D. W. L., Kim, J. H., Mount, H. T. J., & Bazinet, R. P. (2008). The low density lipoprotein receptor is not necessary for maintaining mouse brain polyunsaturated fatty acid concentrations. *Journal of Lipid Research*, *49*(1), 147–152. <https://doi.org/10.1194/jlr.M700386-JLR200>
- Clandinin, M. T., Chappell, J. E., Leong, S., Heim, T., Swyer, P. R., & Chance, G. W. (1980a). Extrauterine fatty acid accretion in infant brain: Implications for fatty acid requirements. *Early Human Development*, *4*(2), 131–138. [https://doi.org/10.1016/0378-3782\(80\)90016-x](https://doi.org/10.1016/0378-3782(80)90016-x)

- Clandinin, M. T., Chappell, J. E., Leong, S., Heim, T., Swyer, P. R., & Chance, G. W. (1980b). Intrauterine fatty acid accretion rates in human brain: Implications for fatty acid requirements. *Early Human Development*, *4*(2), 121–129. [https://doi.org/10.1016/0378-3782\(80\)90015-8](https://doi.org/10.1016/0378-3782(80)90015-8)
- Coplen, T. B. (1995). Discontinuance of SMOW and PDB. *Nature*, *375*(6529), Article 6529. <https://doi.org/10.1038/375285a0>
- Coplen, T. B., Brand, W. A., Gehre, M., Gröning, M., Meijer, H. A. J., Toman, B., & Verkouteren, R. M. (2006a). New guidelines for delta13C measurements. *Analytical Chemistry*, *78*(7), 2439–2441. <https://doi.org/10.1021/ac052027c>
- Coplen, T. B., Brand, W. A., Gehre, M., Gröning, M., Meijer, H. A. J., Toman, B., & Verkouteren, R. M. (2006b). New guidelines for delta13C measurements. *Analytical Chemistry*, *78*(7), 2439–2441. <https://doi.org/10.1021/ac052027c>
- Drickamer, L. C. (1990). Seasonal variation in fertility, fecundity and litter sex ratio in laboratory and wild stocks of house mice (*Mus domesticus*). *Laboratory Animal Science*, *40*(3), 284–288.
- Eiler, J., Cesar, J., Chimiak, L., Dallas, B., Grice, K., Griep-Raming, J., Juchelka, D., Kitchen, N., Lloyd, M., Makarov, A., Robins, R., & Schwieters, J. (2017). Analysis of molecular isotopic structures at high precision and accuracy by Orbitrap mass spectrometry. *International Journal of Mass Spectrometry*, *422*, 126–142. <https://doi.org/10.1016/j.ijms.2017.10.002>
- Fu, X., Deja, S., Fletcher, J. A., Anderson, N. N., Mizerska, M., Vale, G., Browning, J. D., Horton, J. D., McDonald, J. G., Mitsche, M. A., & Burgess, S. C. (2021). Measurement of lipogenic flux by deuterium resolved mass spectrometry. *Nature Communications*, *12*(1), Article 1. <https://doi.org/10.1038/s41467-021-23958-4>
- Giriko, C. Á., Andreoli, C. A., Mennitti, L. V., Hosoume, L. F., Souto, T. D. S., Silva, A. V. da, & Mendes-da-Silva, C. (2013). Delayed physical and neurobehavioral development and increased aggressive and depression-like behaviors in the rat offspring of dams fed a high-fat diet. *International*

- Journal of Developmental Neuroscience: The Official Journal of the International Society for Developmental Neuroscience*, 31(8), 731–739. <https://doi.org/10.1016/j.ijdevneu.2013.09.001>
- Giuliano, V., Lacombe, R. J. S., Hopperton, K. E., & Bazinet, R. P. (2018). Applying stable carbon isotopic analysis at the natural abundance level to determine the origin of docosahexaenoic acid in the brain of the fat-1 mouse. *Biochimica Et Biophysica Acta. Molecular and Cell Biology of Lipids*, 1863(11), 1388–1398. <https://doi.org/10.1016/j.bbaliip.2018.07.014>
- Heyser, C. J. (2003). Assessment of Developmental Milestones in Rodents. *Current Protocols in Neuroscience*, 25(1), 8.18.1-8.18.15. <https://doi.org/10.1002/0471142301.ns0818s25>
- Hoffman, D. W., & Rasmussen, C. (2022). Absolute Carbon Stable Isotope Ratio in the Vienna Peedee Belemnite Isotope Reference Determined by ¹H NMR Spectroscopy. *Analytical Chemistry*, 94(13), 5240–5247. <https://doi.org/10.1021/acs.analchem.1c04565>
- Hopperton, K. E., Trépanier, M.-O., Giuliano, V., & Bazinet, R. P. (2016). Brain omega-3 polyunsaturated fatty acids modulate microglia cell number and morphology in response to intracerebroventricular amyloid- β 1-40 in mice. *Journal of Neuroinflammation*, 13(1), 257. <https://doi.org/10.1186/s12974-016-0721-5>
- Horton, J. D., Goldstein, J. L., & Brown, M. S. (2002). SREBPs: Activators of the complete program of cholesterol and fatty acid synthesis in the liver. *The Journal of Clinical Investigation*, 109(9), 1125–1131. <https://doi.org/10.1172/JCI15593>
- Kim, J., Frey, W., He, H., Kim, H., Ekram, M., Bakshi, A., Faisal, M., Perera, B., Ye, A., & Teruyama, R. (2014). Peg3 Mutational Effects on Reproduction and Placenta-Specific Gene Families. *PLoS One*, 9. <https://doi.org/10.1371/annotation/8de03ec5-e62c-4135-87ee-4928475090a8>
- Lacombe, R. J. S., & Bazinet, R. P. (2020). Natural abundance carbon isotope ratio analysis and its application in the study of diet and metabolism. *Nutrition Reviews*. <https://doi.org/10.1093/nutrit/nuaa109>

- Lacombe, R. J. S., & Bazinet, R. P. (2021). Natural abundance carbon isotope ratio analysis and its application in the study of diet and metabolism. *Nutrition Reviews*, 79(8), 869–888.
<https://doi.org/10.1093/nutrit/nuaa109>
- Lacombe, R. J. S., Giuliano, V., Chouinard-Watkins, R., & Bazinet, R. P. (2018). Natural Abundance Carbon Isotopic Analysis Indicates the Equal Contribution of Local Synthesis and Plasma Uptake to Palmitate Levels in the Mouse Brain. *Lipids*, 53(5), 481–490. <https://doi.org/10.1002/lipd.12046>
- McKINNEY, C. R., McCREA, J. M., Epstein, S., Allen, H. A., & Urey, H. C. (1950). Improvements in mass spectrometers for the measurement of small differences in isotope abundance ratios. *The Review of Scientific Instruments*, 21(8), 724–730. <https://doi.org/10.1063/1.1745698>
- Murphy, E. J. (2006). Stable isotope methods for the in vivo measurement of lipogenesis and triglyceride metabolism. *Journal of Animal Science*, 84 Suppl, E94-104.
https://doi.org/10.2527/2006.8413_supple94x
- Naudí, A., Cabré, R., Dominguez-Gonzalez, M., Ayala, V., Jové, M., Mota-Martorell, N., Piñol-Ripoll, G., Gil-Villar, M. P., Rué, M., Portero-Otín, M., Ferrer, I., & Pamplona, R. (2017). Region-specific vulnerability to lipid peroxidation and evidence of neuronal mechanisms for polyunsaturated fatty acid biosynthesis in the healthy adult human central nervous system. *Biochimica Et Biophysica Acta. Molecular and Cell Biology of Lipids*, 1862(5), 485–495.
<https://doi.org/10.1016/j.bbalip.2017.02.001>
- Newell, D. B., & Tiesinga, E. (2019). *The international system of units (SI): 2019 edition* (NIST SP 330-2019; p. NIST SP 330-2019). National Institute of Standards and Technology.
<https://doi.org/10.6028/NIST.SP.330-2019>
- Orso, R., Wearick-Silva, L. E., Creutzberg, K. C., Centeno-Silva, A., Glusman Roithmann, L., Pazzin, R., Tractenberg, S. G., Benetti, F., & Grassi-Oliveira, R. (2018). Maternal behavior of the mouse dam

- toward pups: Implications for maternal separation model of early life stress. *Stress*, 21(1), 19–27. <https://doi.org/10.1080/10253890.2017.1389883>
- Ortega-Prieto, P., & Postic, C. (2019). Carbohydrate Sensing Through the Transcription Factor ChREBP. *Frontiers in Genetics*, 10. <https://www.frontiersin.org/articles/10.3389/fgene.2019.00472>
- Rosenfeld, C. S., Grimm, K. M., Livingston, K. A., Brokman, A. M., Lamberson, W. E., & Roberts, R. M. (2003). Striking variation in the sex ratio of pups born to mice according to whether maternal diet is high in fat or carbohydrate. *Proceedings of the National Academy of Sciences of the United States of America*, 100(8), 4628–4632. <https://doi.org/10.1073/pnas.0330808100>
- Schimmelmann, A., Qi, H., Coplen, T. B., Brand, W. A., Fong, J., Meier-Augenstein, W., Kemp, H. F., Toman, B., Ackermann, A., Assonov, S., Aerts-Bijma, A. T., Brejcha, R., Chikaraishi, Y., Darwish, T., Elsner, M., Gehre, M., Geilmann, H., Gröning, M., Hélie, J.-F., ... Werner, R. A. (2016). Organic Reference Materials for Hydrogen, Carbon, and Nitrogen Stable Isotope-Ratio Measurements: Caffeines, n-Alkanes, Fatty Acid Methyl Esters, Glycines, l-Valines, Polyethylenes, and Oils. *Analytical Chemistry*, 88(8), 4294–4302. <https://doi.org/10.1021/acs.analchem.5b04392>
- Smith, M. E., Cisbani, G., Metherel, A. H., & Bazinet, R. P. (2022). The majority of brain palmitic acid is maintained by lipogenesis from dietary sugars and is augmented in mice fed low palmitic acid levels from birth. *Journal of Neurochemistry*, 161(2), 112–128. <https://doi.org/10.1111/jnc.15539>
- Smith, S., Gagné, H. T., Pitelka, D. R., & Abraham, S. (1969). The effect of dietary fat on lipogenesis in mammary gland and liver from lactating and virgin mice. *The Biochemical Journal*, 115(4), 807–815. <https://doi.org/10.1042/bj1150807>
- Souto, T. D. S., Nakao, F. S. N., Giriko, C. Á., Dias, C. T., Cheberle, A. I. do P., Lambertucci, R. H., & Mendes-da-Silva, C. (2020). Lard-rich and canola oil-rich high-fat diets during pregnancy

- promote rats' offspring neurodevelopmental delay and behavioral disorders. *Physiology & Behavior*, *213*, 112722. <https://doi.org/10.1016/j.physbeh.2019.112722>
- Teo, J. D., Morris, M. J., & Jones, N. M. (2017). Maternal obesity increases inflammation and exacerbates damage following neonatal hypoxic-ischaemic brain injury in rats. *Brain, Behavior, and Immunity*, *63*, 186–196. <https://doi.org/10.1016/j.bbi.2016.10.010>
- Turgeon, B., & Meloche, S. (2009). Interpreting Neonatal Lethal Phenotypes in Mouse Mutants: Insights Into Gene Function and Human Diseases. *Physiological Reviews*, *89*, 1–26. <https://doi.org/10.1152/physrev.00040.2007>
- Wallace, M., & Metallo, C. M. (2020). Tracing insights into de novo lipogenesis in liver and adipose tissues. *Seminars in Cell & Developmental Biology*, *108*, 65–71. <https://doi.org/10.1016/j.semcdb.2020.02.012>
- Williams, D. R., Carlsson, R., & Bürkner, P.-C. (2017). Between-litter variation in developmental studies of hormones and behavior: Inflated false positives and diminished power. *Frontiers in Neuroendocrinology*, *47*, 154–166. <https://doi.org/10.1016/j.yfrne.2017.08.003>
- Yoon, J. H., Seo, Y., Jo, Y. S., Lee, S., Cho, E., Cazenave-Gassiot, A., Shin, Y.-S., Moon, M. H., An, H. J., Wenk, M. R., & Suh, P.-G. (2022). Brain lipidomics: From functional landscape to clinical significance. *Science Advances*, *8*(37), eadc9317. <https://doi.org/10.1126/sciadv.adc9317>

REVIEWERS' COMMENTS

Reviewer #1 (Remarks to the Author):

This reviewer was impressed by the authors' thorough revision endeavors. I'd like to congratulate the authors for this interesting and important work and happy to support the publication in Nature Communications.

Reviewer #2 (Remarks to the Author):

All my comments were addressed thoroughly and the manuscript was adapted to my satisfaction.

Reviewer #3 (Remarks to the Author):

The authors have addressed all my comments and suggestions successfully. The same was done with the observations and requests done by the other 2 reviewers.

The authors have provided new supporting data, created new supplemental figures, and clarified the language and statistics as we have all requested. The new information increases the clarity of their manuscript and does a good job of explaining the results and conclusions.